# An Immunoinformatics Approach for SARS-CoV-2 in Latam Populations and Multi-Epitope Vaccine Candidate Directed towards the World’s Population

**DOI:** 10.3390/vaccines9060581

**Published:** 2021-06-01

**Authors:** Andrés Felipe Cuspoca, Laura Lorena Díaz, Alvaro Fernando Acosta, Marcela Katherine Peñaloza, Yardany Rafael Méndez, Diana Carolina Clavijo, Juvenal Yosa Reyes

**Affiliations:** 1Grupo de Investigación en Epidemiología Clínica de Colombia (GRECO), Universidad Pedagógica y Tecnológica de Colombia, Tunja 150003, Colombia; greco@uptc.edu.co (A.F.C.); laura.diaz06@uptc.edu.co (L.L.D.); alvaro.acosta@uptc.edu.co (A.F.A.); marcela.penaloza@uptc.edu.co (M.K.P.); yardany.mendez@uptc.edu.co (Y.R.M.); 2Facultad de Ingeniería y Ciencias, Pontificia Universidad Javeriana Cali, Santiago de Cali 760031, Colombia; diana.clavijo@javerianacali.edu.co; 3Laboratorio de Simulación Molecular, Facultad de Ciencias Básicas y Biomédicas, Universidad Simón Bolívar, Barranquilla 080002, Colombia

**Keywords:** SARS-CoV-2, vaccine, LATAM, in silico

## Abstract

The coronavirus pandemic is a major public health crisis affecting global health systems with dire socioeconomic consequences, especially in vulnerable regions such as Latin America (LATAM). There is an urgent need for a vaccine to help control contagion, reduce mortality and alleviate social costs. In this study, we propose a rational multi-epitope candidate vaccine against SARS-CoV-2. Using bioinformatics, we constructed a library of potential vaccine peptides, based on the affinity of the most common major human histocompatibility complex (HLA) I and II molecules in the LATAM population to predict immunological complexes among antigenic, non-toxic and non-allergenic peptides extracted from the conserved regions of 92 proteomes. Although HLA-C, had the greatest antigenic peptide capacity from SARS-CoV-2, HLA-B and HLA-A, could be more relevant based on COVID-19 risk of infection in LATAM countries. We also used three-dimensional structures of SARS-CoV-2 proteins to identify potential regions for antibody production. The best HLA-I and II predictions (with increased coverage in common alleles and regions evoking B lymphocyte responses) were grouped into an optimized final multi-epitope construct containing the adjuvants Beta defensin-3, TpD, and PADRE, which are recognized for invoking a safe and specific immune response. Finally, we used Molecular Dynamics to identify the multi-epitope construct which may be a stable target for TLR-4/MD-2. This would prove to be safe and provide the physicochemical requirements for conducting experimental tests around the world.

## 1. Introduction

Severe acute respiratory syndrome coronavirus 2 (SARS-CoV-2), a pathogen that emerged towards the end of 2019, primarily affects the respiratory tract. It is transmitted from person to person via respiratory droplets, aerosols containing viral particles, and direct contact of the mucosa with contaminated surfaces [1,2].

The first infected patient was identified in Wuhan, Hubei Province, China. The origin of the virus is thought to be the Wuhan seafood market, although some cases had no connection to this location. The virus spread rapidly through Wuhan and shortly thereafter to the rest of China’s provinces [1,3]. By 20 February 2020, 19 countries had reported cases and mortalities caused by coronavirus disease 2019 (COVID-19). In March of 2020, the World Health Organization (WHO) declared SARS-CoV-2 the etiologic agent of the first pandemic caused by a coronavirus [3,4].

In Latin America (LATAM), COVID-19 was first reported in Sao Paulo, Brazil, on 25 February 2020 with the case of a 61-year-old male who has traveled to Italy [5]. Subsequently, cases were reported in other LATAM countries, including Chile [6] and Colombia [7]. Haiti was the last LATAM country to report the arrival of COVID-19 on 19 March 2020. The arrival of COVID-19 in LATAM presents a great challenge to a healthcare infrastructure that is susceptible to problems, such as the lack of SARS-CoV-2 testing, personal protective equipment, and intensive care unit beds. Additionally, mass migration facilitates contact and spread across borders [8]. As the pandemic progressed, a lack of effective control measures accelerated infection and death rates in comparison to other European countries [9,10]. We believe that research institutions in LATAM should come together in the development of a universal vaccine for four fundamental reasons: (1) SARS-CoV-2 has caused high morbidity and mortality worldwide; (2) it is highly contagious; (3) countries, including those in LATAM, were not prepared for the pandemic, and (4) genomic variation could occur during the pandemic, and changes to antigenic sites in vaccine formulations may be required [5,11,12,13]. By the end of the first week of November 2020, about 60 million cases were seen worldwide, with 1.2 million deaths globally [14].

COVID-19 has a broad range of clinical manifestations in patients ranging from asymptomatic to acute respiratory distress syndrome that may lead to death. A greater risk of mortality in patients occurs in unison with comorbidities, such as arterial hypertension, chronic obstructive pulmonary disease, diabetes, and vascular diseases, especially cerebrovascular disease [4,15].

Studies of similar pathologies, such as severe acute respiratory syndrome (SARS), have shown that the independent expression of type 1 interferon (IFN-1)-stimulated genes and Toll-like receptor (TLR) 3 and 4 are associated with better outcomes in infected rats [16]. Also, individuals with a homozygous expression of the polymorphic variants of L-SIGN are known to have a better viral binding capacity, increased viral degradation, and diminished cell to cell infection [17]. The expression of the HLA-C*15:02 and HLA-DR*03:01 alleles has also been associated with viral clearance. These alleles facilitate viral antigen presentation and, consequently, the elimination of SARS-CoV mediated by T lymphocytes (TLs) CD8+/CD4+ and natural killer cells [18].

From these findings, we can infer that the preservation of IFN-1 production and some alleles of HLA-I and HLA-II may be related to asymptomatic conditions and mild symptoms in COVID-19 patients [18]. In contrast, changes to IFN-1-producing signal pathways (such as polymorphisms or mutations) that compromise a patient’s innate immunity, and differential expression of sex-dependent angiotensin-converting enzyme 2 (ACE2) receptor, may be associated with non-modifiable risk factors [19].

### 1.1. Virology of SARS-CoV-2

SARS-CoV-2 belongs to the Coronaviridae family and is part of the β group of coronaviruses. It is a 29.9 kb, positive-sense, single-stranded, enveloped RNA virus. It is similar to the etiologic agents that cause SARS and Middle East Respiratory Syndrome (MERS), which share 79.5% and 50% of their identity with SARS-CoV-2, respectively [3,20]. Coronavirus genomes are composed of 6–11 open reading frames (ORFs) [21], with the first ORF (ORF1a/b) containing two-thirds of the viral RNA. This ORF translates the pp1a and pp1ab polyproteins and encodes 16 non-structural proteins (NSPs). The other ORFs encode structural and accessory proteins. The genome contains accessory genes: two between the spike surface glycoprotein (SP) and small envelope protein genes (ORF3a and 3b); four between the matrix protein (MG) and nucleocapsid protein (NP) genes (6, 7a, 7b, 8), 9a, and 9b in the NP gene; and 10 after the NP gene [3,22,23]. The genome structure is shown in (Appendix A).

### 1.2. Medication for COVID-19

Currently, only Dexamethasone and Remdesivir have been shown to modify the natural course of the disease. The “RECOVERY” randomized clinical trial (RCT) was carried out in England. In this study, oral or intravenous Dexamethasone 6 mg/day was associated with a reduction in the mortality of patients requiring assisted ventilation and patients who were in the second week of clinical diagnosis [24]. Another study, “ACTT1”, involved 10 countries in a multicenter RCT. The trial showed that the nucleotide analogue prodrug Remdesivir, which used a loading dose of 200 mg and a maintenance dose of 100 mg/day, significantly reduced the time it took for patients to recover from COVID-19. Additionally, the need for supplemental oxygen after 10 days of intravenous treatment diminished [25]. However, the results of the WHO multicenter RCT “SOLIDARITY” study on the evaluation of the effect of “off label” drugs, which included the follow-up of about 11,330 adults from 4 continents, 30 countries and 405 hospitals, did not find Remdesivir to be associated with substantial prevention of in-hospital mortality, benefiting only a small fraction of patients when using a loading dose and standard maintenance [26]. Currently, the Food and Drug Administration has approved the use of Remdesivir and Dexamethasone for COVID-19, although the latter is only a temporary approval [25,27]. Other therapies have not shown improvements in mortality or recovery in patients with COVID-19. In an RCT carried out in China, patients with severe or life-threatening COVID-19 symptoms were studied. The findings showed that convalescent plasma did not result in clinical improvement after 28 days of follow-up, demonstrating no superiority to standard medical management [28]. In India, these findings were subsequently confirmed in a national multicenter RCT titled “PLACID” [29]. This is consistent with the “TSUNAMI” RCT, recently published from the Italian population with COVID-19, where it was not evidenced that a high antibody titer present in convalescent plasma will result in a reduction in mortality at 30 days, nor less need for invasive mechanical ventilation, indicating only a marginal effect in the absence of acute respiratory distress [30]. Other “off label” drugs are being evaluated worldwide to assess their usefulness against COVID-19. Tocilizumab, an IL-6 receptor antagonist, is being tested in an RCT (EudraCT: No. 2020-001408-41) in Germany to evaluate its efficacy and safety in patients with severe COVID-19 pneumonia. In the phase III RCT results, the use of Canakinumab, an IL-1β inhibitor, evidence to reduce the days of the hospitalization and mortality, in patients who do not require invasive ventilatory assistance, but do require supplemental O2 [31]. On the other hand, the use of Sarilumab, an IL-6 receptor antagonist, in patients with severity criteria, no benefit was observed compared to placebo [32]. Another phase II RCTs involving Thalidomide (NCT04273529), and monoclonal antibodies such as Anakinra (NCT04603742), Gimsilumab (NCT04351243), and Ruxolitinib are ongoing (NCT04359290).

The authors hypothesize that it could slow the progression of pneumonia and inflammation induced by SARS-CoV-2 [33]. On the other hand, promising results have been observed using mesenchymal cells derived from human umbilical cords [34] in patients with moderate and severe COVID in China. This has resulted in complete resolution of pneumonia within two weeks of infection without mortality and serious adverse reactions. It is currently in a non-randomized, controlled trial (NCT04288102).

### 1.3. Vaccine for SARS-CoV-2

A vaccine will be the most cost-effective strategy for preventing infection and reducing COVID-19-related morbidity and mortality [35]. According to a WHO report, vaccine studies for COVID-19 are using different vaccine strategies, such as non-replicating viral vectors, RNA, inactivated, viral-like particles, protein Subunit, peptides, and DNA approaches. To date, there are more than 100 vaccine candidates in preclinical trials [36]. Vaccine strategies and their phases are summarized in (Appendix A). In both SARS-CoV and SARS-CoV-2, several candidates structural proteins have been studied as vaccine targets. SP has been widely studied due to their capacity to induce neutralizing antibodies (NAb) that prevent the virus from binding and fusing with ACE2 receptor, as well as activating TLs. Various vaccine models have been proposed using the complete protein, SP binding domain, virus-like particles, DNA, or viral vectors [37,38]. NP was found to be expressed in large amounts during infection. NP is highly immunogenic and is a potential vaccine target. It develops a memory response from TLs that remain up to 11 years after SARS-CoV infection, detecting at the same time TLs of a specific memory, in addition to NP towards other structural proteins such as SP and MG [39]. NP has recently been identified as an important virulence trait in the clinical outcome of COVID-19. In patients with moderate and severe COVID-19, sC5b-9, C4d, and NP are associated with respiratory failure (PAFI < 40 kPa) and the need for oxygen therapy. In murine models, this could be explained because NP is capable of dimerizing, activating, and cleaving MASP2 [40], a serine protease that can induce complement activation [41]. NP, MBL, MASP-2, C4 alpha, and C3 or C5b-9 have been observed in the lung tissue of patients with fatal COVID-19, which could indicate new therapeutic targets [40]. Previously, in a murine model infected by SARS-CoV and their principal structural proteins, NP alone resulted in a dysregulated inflammatory response in addition to pulmonary infiltrate of neutrophils, eosinophils, and lymphocytes. This led to a thickening of the pulmonary epithelium and severe pneumonia when using NP or SARS-CoV but not SP [42]. Therefore, the full-length protein NP as a vaccine target for SARS-CoV-2 could be considered less safe when compared to SP, since they share a homology greater than 90% [43]. Consequently, if it is used as a vaccine target, shorter length approaches should be chosen, since immunogenic peptides of NP have been found capable of evoking an immune response and inhibiting viral replication in influenza A and SARS-COV viruses [43,44,45,46,47].

Other experimental studies about structural proteins on SARS-CoV have identified the SP and MG as targets to induce NAb in the serum of patients [48]. Furthermore, in SARS-CoV-2, IgG-like NAb, which targets the receptor-binding domain (RBD) of SP, has been observed in patients with severe COVID-19 and convalescent patients [49,50]. NAb of NP has been related to the number of specific TLs which produce IFN-γ [50]. In SARS-CoV, NP, MG, and E have been shown not to produce NAb; however, they are potential antigens for antiviral cytotoxic T cells (CTL) [51]. Taking into account the above, SP, NP, and MG proteins are the main targets for stimulating the creation of antibodies through B lymphocytes (BLs) and TLs response when using specific peptides and incomplete proteins. Theoretically, individuals without previous infection by SARS or COVID-19 have shown specific TLs capable of recognizing peptides from SARS-CoV-2 polyprotein 1 a/b that could stop the viral cycle at an early stage, thus avoiding an expansive immune effect [52]. Although the impact of these findings is unknown, studies derived from other coronaviruses could indicate greater protection from a worse clinical outcome given by less severity of the disease [53]. However, a rational vaccine approach could include: (1) neutralization based on structural proteins, (2) early detection of replication-related proteins, and (3) accessory proteins involved in the formation of the mature virion, as well as protection against the residual damage in NP.

In LATAM, the outbreak has had negative sociocultural, economic, and political effects. Without a doubt, everyone will be affected by further outbreaks, especially vulnerable populations. The construction of a vaccine is therefore imperative for reducing health costs and morbimortality associated with COVID-19 [5,11,12,13].

Using an immunoinformatics approach, we propose a multi-epitope peptide vaccine model based on peptide binding properties extracted from conserved regions of SARS-CoV-2 proteomes. Moreover, we provide considerations for binding these peptides with HLA-I and II alleles most frequently found in LATAM (Figure 1).

## 2. Materials and Methods

### 2.1. SARS-CoV-2 Proteome Recovery

The proteomes of 92 SARS-CoV-2 strains were downloaded from the Assembly database available at the National Center for Biotechnology Information (NCBI) (https://www.ncbi.nlm.nih.gov/assembly, accessed on 27 April 2020). The metadata was extracted manually. Origin, isolation type, and sequencing date were identified. The interactions between SARS-CoV-2 proteins and the other factors studied were illustrated using the freely available web resource BioRender.com.

### 2.2. Identification of Amino Acid Sequences Conserved in SARS-CoV-2 Proteomes

To exclude the potential non-synonymous substitutions in the amino acid sequences from the recovered SARS-CoV-2 proteomes, we performed multiple alignments using the MAFFT server v7.0 (https://mafft.cbrc.jp/alignment/server/, accessed on 29 April 2020) [54], with the genome NC_04551 as a reference. This contained structural and accessory proteins, including the transcript ORF1a/b. The resulting preserved sequences formed non-redundant sequences which were used for further analysis.

### 2.3. Identification of the Most Frequent HLA-I and HLA-II Alleles in LATAM Population

The allele frequencies for each HLA-I (HLA-A, HLA- B and HLA-C) and HLA-II (HLA-DRB) allele for the Central and South American regions were downloaded using a specialist search tool (http://www.allelefrequencies.net/default.asp, accessed on 10 May 2020). The phenotype percentage (PF) was calculated using Equation (Equation 1). The most frequent phenotypes were separated. PF values greater than the Me were grouped by locus and country. Tableau v2020.2 [55] was used to create a geographic representation of the most frequent alleles.
(1)$$PF=1-{\left(1-AlleleFrequency\right)}^{2}$$

### 2.4. HLA-I and HLA-II Allele Selection

Only the alleles corresponding to the DRB loci were identified for HLA-II analysis due to the higher precision of these predictions to identify individual training and cross-training with other HLA-II molecules in algorithms based on artificial neural networks (such as NetMHCII and NetMHCIIpan). In general, HLA-I algorithms had a greater number of trained molecules and produced more accurate results than those of HLA-II. DRB is highly diverse in nature: >3000 isolates have been identified in humans [56]. This variability seems to be caused by the rapid and diverse evolutionary response of extracellular pathogens [57].

The HLA-DQ loci were not considered due to their association with autoimmunity and the instability of the complexes with their own epitopes. Additionally, the HLA-DP loci were not taken into account due to their less frequent associations with pathogens, which primarily contribute to tolerance and the proper function of innate virus response [58].

For HLA-I, the individual alleles most frequently found in LATAM at loci A, B, and C were used instead of identifying the promiscuous alleles more likely to identify SARS-CoV-2 proteins because the latter approach may lead to reduced precision when searching for an effective vaccine. The HLA-I group of genes is also isolated more frequently than those in DRB, with the B locus being the most diverse.

### 2.5. Prediction of T Helper Lymphocytes (HTL) Epitope

To predict TLs and CD4+ epitopes, we used conserved sequences of SARS-CoV-2 structural proteins with at least 15 mers. The algorithm NetMHCII v2.3 was used. This approach is based on training individual molecules from complex experiments, which gives greater accuracy [59]. The molecules available to make an agglutination prediction were as follows: DRB1_0101,DRB1_0103 DRB1_0301,DRB1_0401,DRB1_0402,DRB1_0403, DRB1_0404, DRB1_0405, DRB1_0701, DRB1_0801, DRB1_0802, DRB1_0901, DRB1_1001, DRB1_1101, DRB1_1201, DRB1_1301, DRB1_1302, DRB1_1501 and DRB1_1602. For each allele, all predictions were grouped and then filtered by <2% rank to identify probable binders that had a greater number of predictions with <IC50 affinities. The predictions that remained after filtering were considered to be potential HLA-II epitopes capable of being recognized by HTL CD4+.

### 2.6. Prediction of the CTL Epitope

To break through the SARS-CoV-2 viral assembly at an early stage of infection, we used structural accessories, and NSPs to predict potential epitopes for CTLs. We predicted peptides related to HLA molecules most common in LATAM, using the sequences of proteomes with at least 9 mers.

The two algorithms with the best experimental correlations were used, NetMHCpan and MHCflurry [60,61]. These approaches used cross-trained and individually trained neural networks. These approaches are based on experimental data from the Immune Epitope Database and Analysis Resource (IEBD) database, integration of eluted peptides, and other ligands identified by mass spectrometry.

A portable version of NetMHCpan 4.0 was used following request to the “Health Tech” area of the Technical University of Denmark (https://services.healthtech.d-tu.dk/cgi-bin/sw_request, accessed on 3 May 2020). MHCflurry v1.6.1 was downloaded from the author’s github repository (https://github.com/openvax/mhcflurry/releases, accessed on 3 May 2020). The results were filtered using a strong binding prediction (<2% rank) based on the author’s recommendations and using a methodology based on binding affinity. In MHCflurry, the sequences were filtered with an affinity identity percentile <2 and cutoff values for predicting bond strength of up to 100 nM.

The peptide sequences resulting from the algorithms were tested for their immunogenicity using an immunogenicity tool that uses the position of the residues in the HLA molecule cleft and characteristics such as their basic nature and size to predict the interaction with the CD8+ CTL receptor and the initiation of an immunogenic response. Epitopes capable of generating an immune response by CTLs were grouped by their positive score. Those with positive values were considered candidates for further evaluation of antigenicity, allergenicity, and toxicity [62].

### 2.7. Antigenicity Prediction

The intrinsic characteristics of antigens, such as the protein nature, structure, physicochemical and extrinsic properties, are known to be related to immune response and are largely regulated by HLA, self-tolerance and host genetics [63]. The Vaxijen 2.0 server (http://www.ddg-harmfac.net/vaxijen/VaxiJen/VaxiJen.html, accessed on 19 May 2020) was used to identify non-redundant epitopes predicted to bind strongly to HLA-I and II molecules with antigenic potential. This server uses an antigen reference database and compares the physicochemical properties of the amino acids by cross-covariance. It converts the sequences of non-redundant epitopes that result in strong agglutinations into uniform vectors [64]. This process discriminates between probable and unlikely antigens. Peptides with strong agglutinations and an antigenicity threshold of >0.4 were considered to be potential immunogens.

### 2.8. Prediction of Allergenicity and Toxicity

To rule out probable allergic and toxic reactions caused by the interaction of peptides, we used AllerTOP v. 2.0. (https://www.ddg-pharmfac.net/AllerTOP/, accessed on 19 May 2020) [65] and ToxinPred (http://crdd.osdd.net/raghava/toxinpred/, accessed on 19 May 2020) [66]. AllerTOP employs a similar approach to Vaxijen 2.0: it uses an auto-covariance transformation to normalize the alignment of peptides with immunogenic potential and includes an automatic and manual pull of cured allergenic and non-allergenic proteins. Probable allergens are determined by several automatic machine learning techniques. These methods have a sensitivity and specificity of 0.87 and 0.90, respectively, which is achieved using descriptors of the allergy-related characteristics of individual amino acids [61]. ToxinPred uses a machine vector support technique to discriminate, via amino acid composition analysis and a quantitative matrix, the peptides with immunogenic potential and toxic probability. The position and frequency were analyzed along with other characteristics of the amino acids that are most abundant in toxic peptides to obtain accuracy close to 97%. Non-toxic and non-allergenic peptides with immunogenic potential were considered to be potential SARS-CoV-2 vaccine targets.

### 2.9. Allelic Promiscuity and Identification of Experimental Epitopes

Matrices containing potential vaccine epitopes of HLA-I and HLA-II molecules were created to identify the promiscuity of each of the predictions with vaccine potential. They were characterized as follows: peptides conserved in proteomes of 9 or 15 mer, immunogenic characteristics, non-toxic and non-allergenic, and a strong affinity to the most common alleles found in LATAM. These matrices were grouped by allelic HLA class to identify possible non-redundant combinations capable of covering all the HLA molecules tested. Peptides had equal predictions, the highest score from Vaxijen was used to decide the best peptide candidate. The groups with the lowest number of peptides were considered to be optimal for experimental validation as part of the rational multi-epitope construct.

Given that experimentally validated peptides in closely related viruses such as SARS-CoV could also be immunogenic targets, the peptides comprising the multi-epitope construct were subject to a 90% BLAST search in the Immune Epitope Database Analysis Resource (IEDB; https://www.iedb.org/, accessed on 3 November 2020).

### 2.10. Flexible Peptide-Protein Docking

We estimated the conformation of complexes between peptides with vaccine potential and the HLA molecules found most frequently in LATAM. These must interact as a flexible and stable anchor that allows interaction with CD4+ and CD8+ TLs receptors [67]. The CABS coupling algorithm was used to perform global molecular coupling between proteins and peptides. This allows full flexibility of the peptide and receptor backbone [68]. A standalone python package of CABS v 0.9.16 was downloaded from a repository (https://bitbucket.org/lcbio/cabsdock/downloads/, accessed on 4 July 2020).

The HLA molecules with experimental resolution found in RCSB PDB were used as the receptors. We identified the corresponding chains, according to the HLA class (α and β-2 globulin for HLA-I; α and β for HLA-II) while predicting the secondary structure of the peptides by PSIPRED through the RPBS web portal (https://bioserv.rpbs.univ-paris-diderot.fr/index.html, accessed on 4 July 2020). The default parameters were used for all other parameters. The energy calculations of the complexes were calculated using the Prodigy server (https://bianca.science.uu.nl/prodigy/, accessed on 4 July 2020) [69].

### 2.11. Prediction of the BL Epitope

To identify potential continuous epitopes capable of stimulating a BL response, we used the artificial neural network-based ABCpred tool [70]. The cutoff threshold was set to 0.90, which allowed for a greater specificity to be selected. To predict potential discontinuous BL epitopes, Discotope v2.0 [71] (server version) (http://www.cbs.dtu.dk/services/DiscoTope/, accessed on 20 July 2020) was used. The cutoff threshold and specificity were set to −2.5 and 80%, respectively.

This prediction included both SP and NP, which were extracted from the RCSB PDB with the following identifiers: 6lzg, 6m0j, 6vw1, 6w41, 6yla, 6yor, 6csb, 6vxx, 6vyb, 6m3m, 6vyo and 6wkp. Predictions resulting from the algorithms were considered to be potential epitopes. The peptide with the highest ABCpred score was also chosen with a partial or total presence in the experimental sequences identified as SARS-CoV immunological targets in IEBD with a 90% BLAST alignment. Also, it was located in potentially immunogenic regions, either because they were shown in Discotope-provided areas from various experimental structures or because they were contained in a specific protein domain. The antigenic peptides with values >0.4 in Vaxijen that were non-toxic and non-allergenic according to AlgPred [72] (http://crdd.osdd.net/raghava/algpred/submission.html, accessed on 25 July 2020) were chosen as potential BL epitopes and were added to the multi-epitope construct.

### 2.12. Recuperation of Validated Epitopes and Identification of Post-Transcriptional Modifications

To analyze the possible post-transcriptional modifications of potential promiscuous vaccine peptides (PPVPs), we used the NetNGlyc 1.0 server [73] (http://www.cbs.dtu.dk/services/NetNGlyc/, accessed on 3 November 2020) with the Wuhan-Hu-1 strain being used as a reference for the accessory proteins and NP. Regarding SP, the structure PDB:6VSB was used for non-structural proteins, which are replicated in the cytoplasm but have some luminal domains that are prone to N-glycosylation. UNIPROT (https://www.uniprot.org, accessed on 3 November 2020) was used to pinpoint their cellular location.

To identify potential immunogenic regions as well as recent validations of the PPVPs, we downloaded the epitope database available on 3 November 2020. It consisted of 822 validated epitopes for TLs and 330 for BLs from SARS-CoV-2 assays. Additionally, we used supplementary assays as targets for the HLA molecule that comes from immunitrack (https://www.immunitrack.com/free-coronavirus-report-for-download/, accessed on 3 November 2020), which consists of a library of peptides tested as binders of various HLA-I and II alleles, taking as a threshold reference stability greater than 30, according to authors’ recommendations.

### 2.13. Multi-Epitope Vaccine Construct Design

To create a multi-epitope vaccine, potential HTL, CTL, and BL epitopes were linked using GPGPG, AAG and KK linkers, respectively. For a better immunogenic response, four adjuvants were added using the EAAAK linker, β-defensin 3, Tetanus toxin peptide and a Diphtheria toxin peptide linked by a cathepsin cleavage site (TpD), and a universal T helper epitope (PADRE). A peptide domain (CTGKSC) targeted by M cells was also added to the C terminal, promoting transcytosis between enterocytes and antigenic uptake at the intestinal level [74].

### 2.14. Analysis of Antigenicity and Allergenicity of the Vaccine Construct

To define whether the final design of the multi-epitope proposal was safe and viable, its antigenic capacity was considered by predicting antigenicity with Vaxijen values with a threshold of >0.4. Its toxicity was predicted using ToxinPred and its allergenicity using Allergen FP [75] (http://ddg-pharmfac.net/AllergenFP/, accessed on 1 December 2020).

### 2.15. Analysis of the Physicochemical Properties of the Multi-Epitope Vaccine Construct

To analyze the construct’s physicochemical properties, the Expasy server and the ProtParam tool [76] were used to determine optimal recognition by the immune system. Negative values for the grand average of hydropathy (GRAVY) [77] and stability are especially important for adequate antigen presentation, as well as determining solubility, which was performed using SolPro. Other parameters related to production and expression were also considered, including the aliphatic index (related to the construct thermostability).

### 2.16. Vaccine Structure Prediction and Validation

The secondary structure of the multi-epitope design was predicted using SOPMA [78] and PSI-PRED [79]. The tertiary structure was predicted using the Robetta server based on homology [80]. An evaluation of the structure quality was carried out using the Ramachandran diagram in the PDBsum [81] and by the ERRAT server [82]. Refinement was performed using 3D Refine [83] and then GalaxyRefine [84].

### 2.17. Molecular Docking of the Multi-Epitope Construct and TLR-4

The TLR-4 receptor was selected as the ideal immunological target of the multi-epitope construct since it is capable of inducing INF production and expressing itself in the cell membrane of dendritic cells. We studied the resulting interactions in the stable formation of the TLR-4/ Myeloid Differentiation Factor 2 (MD-2) complex and the multi-epitope construct using molecular docking, performed using the Cluspro server 2.0 [85] (https://cluspro.bu.edu/login.php, accessed on 10 December 2020). The TLR-4/MD-2 hetero-tetramer was used as a receptor (obtained from PDB RCSB database; ID: 3FXI), and the refined multi-epitope construct was used as a ligand. The residues at the binding interface of the resulting complex were analyzed by PDBsum and plotted by UCSF Chimera v1.14 [86].

### 2.18. Molecular Dynamics Simulation of the Multi-Epitope Construct and TLR-4 Complex

Coordinates of the best model of the multi-epitope construct and TLR-4 were used to perform molecular dynamics analysis. Minimization and molecular dynamic protocols were performed with AMBER 16 [87]. ff14SB force field parameters were used for the amino acid residues [88]. The complex was subjected to unrestricted molecular dynamic simulations for all atoms in an explicit solvent using the PMEMD GPU version algorithm in Amber16 [87].

The Leap module integrated within Amber16 was used to add missing hydrogen atoms and add Cl^−^ ions for neutralization. The systems were immersed in an orthorhombic box using the TIP3P [89] water model. The long-range electrostatic interactions were calculated using the particle mesh Ewald method [90], with a direct space and a vdW cutoff of 12 Å//. An initial minimization was applied to the solute using a potential of 500 kcal mol${}^{-1}$ Å//${}^{2}$ for 10,000 steps using the steepest descent algorithm, followed by 10,000 steps with the conjugate gradient method. Subsequently, 10,000 unrestricted minimization steps were simulated using a conjugate gradient algorithm.

The heating protocol was carried out with a gradual increase in temperature from 0 to 310.15 K using a harmonic restriction of 5 kcal mol${}^{-1}$ Å//${}^{2}$ applied to the solute. A Langevine thermostat with a collision frequency of 1 ps${}^{-1}$ was used with the canonical assembly (NVT). The complex was equilibrated at 310.15 K in an NPT assembly for 10 ns without restriction using the Berendsen barostat to maintain the pressure at 1 bar. The SHAKE [91] algorithm was used to restrict the bonds of all hydrogen atoms. A 2 fs time-step was used with the precision model SPFP [92] in the molecular dynamic simulation.

Finally, 68 ns of production was simulated in an NPT assembly with a target pressure of 1 bar and a pressure coupling constant of 2 ps. Production trajectories were analyzed each 2 ps along the whole simulation using CPPTRAJ and PTRAJ [93].

### 2.19. Codon Optimization and In Silico Cloning

To propose a realistic scenario for peptide vaccine cloning, in silico analyses were conducted to identify the best options for the expression and isolation of the multi-epitope construct. The *Escherichia coli* (*E. coli*) K 12 expression system was selected since it is inexpensive to grow, relatively easy to manipulate genetically, and generally produces high levels of recombinant proteins. Also, it has an optimized lineage for overexpression of recombinant proteins [94]. To identify the best cloning strategy, the complete protein sequence of the multi-epitope construct was first converted into cDNA using Backtranseq reverse translation. The resulting cDNA sequence was further optimized by adapting the most frequently used codons of *E. coli* K 12 to enhance protein expression using the JCAT server [95]. To achieve adequate protein purification, the plasmid vector pET-28a(+) was chosen because of the possibility of labelling polyhistidine towards the N or C terminals of the multi-epitope construct. Therefore, a complete protein was obtained by avoiding possible truncated proteins.

Once the appropriate scheme for cloning was identified, SnapGene v5.1.2 was used to provide enhanced flexibility for displaying and annotating sequences. For this, the HindIII and BamHI restriction sites were used. A cut was made that enabled us to retake a closed structure of the plasmid vector with the appropriate position of the optimized genetic sequence of the construct.

### 2.20. Immune Simulation

The immunogenic behavior of the multi-epitope construct was simulated using the C-IMMSIM server (https://kraken.iac.rm.cnr.it/C-IMMSIM/, accessed on 19 December 2020). This agent-based computer model handles a diverse number of cells representing innate and acquired immunity. By following a set of rules obtained at an experimental level, interactions with the vaccine construct are capable of simulating behaviors that may suggest the probable generation of immune memory. This is achieved by combining the mesoscopic scale of the immune system using three compartments: the bone marrow, thymus, and lymphatic organs. In addition, it uses deep learning tools and molecular level techniques to predict the interaction of the construct and its affinity from the matrices of some HLA molecules. The algorithm can also identify probable linear BLs epitopes from physicochemical parameters. The minimum inter-dose time for current vaccines is no more than 4 weeks. For this reason, simulation values were adjusted with three injections separated by 1, 84, and 168 time-steps, resulting in <4 weeks between doses. The simulation was completed in up to 200 time-steps, with other predetermined values.

To test the response capacity from the interaction between the multi-epitope construction and the immune system, a viral challenge was performed one year after the start of the extended vaccine scheme and was simulated beyond day 460.

## 3. Results

### 3.1. Recovery of SARS-CoV-2 Proteomes

Proteomes from 92 isolates of SARS-CoV-2 were recovered in FASTA format using the GenBank database as a reference. Metadata related to the collection date, city, host, and source of isolation were also downloaded. Isolates were collected by nasopharyngeal (n = 31) and oropharyngeal (n = 12) swabs, and to a lesser extent by bronchoalveolar lavage (n = 11). Sequencing was performed from December 2019 (the Wuhan-Hu-1 strain) to 11 March 2020 (the SARS-CoV-2/human/USA/PC00101P/2020 isolate). Most sequences came from the United States of America (n = 51) and China (n = 27). Only one isolate came from LATAM (Brazil, February 2020). The entire metadata set of proteomes used, as well as access identification, is shown in (Appendix A).

### 3.2. A Vaccine for SARS-CoV-2 Must Take into Account Non-Structural Proteins

“The spike protein acts like an Early Trojan horse.”

SP plays a fundamental role in SARS-CoV-2 infection because it mediates the virus entry into host cells through its S1 domain, which binds to the ACE2 receptor making the subsequent cleavage of the S2 domain.

The increased glycosylation of SP allows it to evade the adaptive immune response and protect epitopes from recognition and antibody neutralization. According to a recent study, antibodies derived from exposure to SP could be related to the clinical outcome by targeting surfactant proteins related to alveolar surfactant [96]. Through viral proteins and RNA receptors, the immune response is the first line of defense against SARS-CoV-2 infection, producing cytokines and IFN-1 [97]. Therefore promoting viral clearance by activating a systemic antiviral state.

As with SARS-CoV, SARS-CoV-2 has proteins encoded by the coronavirus genome that are capable of interfering with the innate immune response subsequently an early phase of infection. This affects various signaling pathways important in maintaining immunity [98] (Appendix A). Opportunely detecting and stopping the pathway interferences could be a more complete approach to vaccination as opposed to only neutralizing RBD or other SP domains. SARS-CoV-2 evidently, has shown that early translated proteins have functions beyond viral replication. They are also capable of deeply and negatively modulating pathways related to IFN, including its synthesis [99].

For example, NSP3 possesses papain-like protease (PLP) domains. Making it is one of the two proteases that participate in the cleavage of NSPs from polyprotein 1 a/b. PLP in SARS-CoV-2 has a more evident dual de-ISGylation capacity than deubiquitination [100]. This impacts pathways up-regulated by IFN, including viral peptide presentation and processing [101].

Although NSP3 is barely evidenced when compared to structural proteins in SARS-CoV and SARS-CoV-2 [102], it is in cohesion with accessory proteins, participating in the assembly of the mature virion [102].

NSP3, and to a lesser extent NSP2 and NSP12, have been identified as the NSPs with the most translation in cell cultures. Other NSPs are associated with the altered transcription and translation of proteins in the host, including IFN [99]. This is achieved by disrupting messenger RNA (NSP16) splicing that selectively blocks translation in ribosomes. This only allows the leading sequence of the viral subgenomic RNA to continue (NSP1) and interrupts the passage of proteins to the membrane. This includes secretory proteins such as IFN, cytokines, and HLA by affecting the SRP complex (NSP8-NSP9, NSP8, NSP9).

Therefore, identifying the mechanisms of action, characteristics of structural NSPs, and accessory transcripts within the process of recognition, entrance, invasion, and replication, is important for finding potential vaccine targets. When given a set of proteins and knowing their mechanism of action inside the cell, identifying the NSPs may be an early approach to avoiding extensive immune involvement. NSPs produced at an early stage and necessary for replication should be considered. Structural protein production is also important since it is essential in the SARS-CoV-2 life cycle.

The frequency and alignment of 92 proteomes were considered to identify conserved regions. This was based on the evidence of the effect of neutralizing antibodies towards structural proteins and the activation/recognition of TL towards structural, non-structural, and accessory proteins from SARS and COVID-19 convalescent patients. Our findings are shown in (Appendix A). The conserved sequence blocks were used in their entirety for the prediction of conserved epitopes. These characteristics are included in our vaccine proposal summarized in (Figure 2).

### 3.3. Most Frequently Identified HLA-I and HLA-II Alleles in LATAM

HLA systems in humans are located in the most polymorphic region of the genome, related to the continuous selection when interacting with extracellular and intracellular pathogens. Furthermore, this diversity corresponds to the rapid adaptation to environmental change, and actually, this has become a hallmark of the first migrations of various human populations [106].

When identifying vaccine targets for SARS-CoV-2 and other emerging pathogens of probable zoonotic origin, it may be possible to generate an immunological response memory. While this would only work in certain populations, given that it is restricted to HLA molecules, it could occur because of the diverse affinities that HLA-I and II molecules have for different sets of peptides that are processed from SARS-CoV-2.

Identifying people and populations with greater susceptibility to COVID-19 could help construct effective healthcare focused on patients who are more susceptible and have an increased risk of mortality. Theoretically, HLA capable of recognizing a greater range of SARS-CoV-2 peptides has been found through computational algorithms, which have been validated in clinical settings, and are associated with better results, especially in heterozygous individuals [107].

Although we lack a precise medicine model that identifies HLA and peptide binding capacity from the SARS-CoV-2 proteome, global initiatives like HLA COVID-19 has been launched to find relevant translational strategies. Additionally, other authors have proposed routine HLA typing in patients with COVID-19 [108].

We identified 168 of the most frequent HLA-I alleles in 17 countries. The HLA-A*02:01 allele was found at an above-average frequency in Argentina, Brazil, Chile, Colombia, Cuba, Ecuador, Nicaragua, Peru, and Venezuela. The alleles A*24:02 and B*40:02 were identified above the 80th percentile; they rank first in frequency in most countries. Some countries had missing data and small sample sizes, e.g., Guatemala (where only the B*53:01 allele was found above the mean) and Trinidad and Tobago (where the only available allele, C*16:02, was used for complementary analyses).

As shown in (Appendix A), the frequencies found between the HLA-I loci were A (n = 126), B (n = 264), and C (n = 64); those alleles above the mean frequencies were A (n = 48), B (n = 91) and C (n = 27).

For HLA-II, none of the countries shared an allele higher than the Median (Me) per country. DRB1*03:01 allele was found with greater frequency in at least 14 countries, except for 3: Mexico, Venezuela, and Paraguay. The HLA-II alleles above the 90th percentile were correlated with most of the allelic groups recovered, except for DRB1*10 and DRB1*12 allelic groups, which were not represented by alleles. DRB1 * 14: 02 was the most frequent allele and was isolated mainly from the Colombian, Brazil, and Argentina populations. The allelic groups and the most frequent alleles found in 18 countries are attached in (Appendix A). All DRB1 alleles listed by NetMHC 3.2, spanning 18 countries, had above average frequencies. (Appendix A) shows the distribution of the main alleles and allelic groups by country according to HLA-I and HLA-II.

### 3.4. Potential Epitopes of TLs, CD4+, CD8+, and BLs Are Contained in the Conserved Sequences of 92 SARS-CoV-2 Proteomes

#### 3.4.1. HLA-II

To identify and select the potential HLA-II epitopes, we used 19 HLA-II molecules from the available DRB gene alleles to perform an agglutination prediction using NetMHCII 2.3. We excluded the conserved sequences that were <15 mers from SP, NP, and MG proteins. In total, 199 peptides with strong binding capacities and antigens were identified. The proteins with the highest number of agglutinations was SP (n = 114), followed by MG (n = 62) and NP (n = 23). The origin of predictions from the conserved sequence and the 199 antigenic peptides are included in (Appendix A).

Appendix A summarizes the conserved sequences present in the SP binding domain of interest. DRB1*01:01, DRB1*04:05, DRB1*10:01, and DRB1*16:02 alleles were found to be promiscuous in this domain (predicted to be strong binders). The allele DRB1*16:02 is found more frequently in LATAM; therefore, it may be related to clinical outcomes and deserves to be studied in more detail.

The 15 predicted mers, with antigenic and strong binding characteristics, were evaluated for allergenic and toxic traits using AllerTOP and ToxinPred, respectively.

#### 3.4.2. HLA-I

To achieve a more precise identification of strong binders from the 167 most frequent HLA-I molecules in LATAM, we used two algorithms based on artificial neural networks identified as the most accurate by IEBD (as of 15 March 2019).

A set of 944 peptides were classified as strong binders and antigens. These were the result of a consensus from the two algorithms. The proteins with the highest number of predictions in order of frequency were as follows: ORF1 transcript (n = 573), SP (n = 163), MG (n = 51), ORF-3a transcript (n = 43), and NP (n = 34). The contribution made by each conserved sequence to these proteins and the binders are included in (Appendix A).

A prediction was not possible for the HLA-B*51:10 allele because it was unavailable in the list. In addition to the antigenicity and strong binding characteristics of these groups, we also considered those with vaccine and immunogenic potential, non-allergenic traits, and non-toxic characteristics on our list. Adaptive immunity protection is now known to change the natural course of the disease by neutralizing SARS-CoV-2 [109].

#### 3.4.3. BLs

The most recent experimental structures of the SP and NP proteins found in RCSB PDB (https://www.rcsb.org/, accessed on 14 July 2020) with the following IDs were used: 6lzg, 6m0j, 6vw1, 6w41, 6yla, 6yor, 6csb, 6vxx, 6vyb, 6m3m, 6vyo, and 6wkp. These structures have special characteristics, such as binding to ACE2 receptor, an antibody extracted from a convalescent SARS patient named CR3022 [110], and conformational pre-fusion in the open and closed state. Analyses were performed using the Discotope 2.0 server adjusted to 80% specificity.

Using the specific SP or NP chains, we implemented the prediction algorithm on all proteins based on their available 3D structure. The predictions, characteristics of the complexes, intervals, and frequency with which probable epitopes were identified in each structure are shown in (Appendix A). For SP, we have indicated three possible continuous epitopes and one discontinuous region that may be important due to the extracellular access. The areas between residues 443–450, 487–494, 496–506, and discontinuous 454–459–460–469–471 seem to be more frequently related to probable interactions with BL. A principal feature of these predictions is the absence of glycosylation in the residues.

For NP, we identified five regions of possible continuous epitopes and two discontinuous ones for the RNA binding domain. On average, the intervals are longer than SP. The regions comprise the intervals in the residues: 59–64, 91–106, 120–130, 136–148, 150–156 and discontinuous between different intervals in the residues 66–82, 115–130, 163–171. The NP and SP with linear and discontinuous epitopes zones described above are illustrated in (Appendix A).

The linear epitope intervals of SP and NP described in the previous structural approach were taken into account, to identify longer sequences (>15 mers), which would be capable of evoking a response mediated by the BL receptor. For this approach, we used the ABCpred server and adjusted specificity to 90%.

Other filters were used to identify two vaccine peptide candidates for BLs, within of SP and NP proteins from SARS-CoV-2, that are presented in (Appendix A). The filters included predicting epitopes in the functional domains using RDB for SP or the RNA binding site for NP, antigenicity, non-toxicity, non-allergenicity, and the identification of conserved sequences in SARS-CoV with experimental immunological evidence available at IEBD. After the filters were applied, the best options were selected to choose the linear epitopes candidates for BL to be incorporated into the multi-epitope construction. These sequences are summarized in (Appendix A).

### 3.5. SARS-CoV-2 MG and SP Contain Conserved Sequences Capable of Interacting with a Large Number of HLA-II Molecules

For HLA-II molecules, a Me of 8 agglutinations was found in the 18 molecules analyzed. The five main alleles that were found to be more capable of recognizing peptides in at least 15 mers were: DRB1*04: 02 (n = 50), DRB1*09: 01 (n = 35), DRB1*08: 01-DRB1*16: 02 (n = 33), DRB1*04: 03 (n = 30), which only appear in their entirety in Chile and Brazil. The DRB1*16: 02 and DRB1*04: 03 alleles were more frequent in the Chile, Brazil, and Colombia populations respectively, which could be related to a better clinical outcome. Although Argentina had these alleles, the frequency in its population is low compared to DRB1*08: 02, which resulted in a lower ability to recognize peptides (n = 13). It should be noted that this allele was found more frequently in other countries such as Brazil, Colombia, and Peru, which could be related to an unfavorable clinical outcome. The alleles with the highest promiscuity in HLA-II molecules were SFRLFARTRSMWSFN (n = 7), a peptide that comes from the MG protein of SARS-CoV-2 and is partially conserved in SARS-CoV. QSIIAYTMSLGAENS (n = 4) and VLSFELLHAPATVCG (n = 4) from the SARS-CoV 2 SP protein followed. The agglutinations between the 199 antigenic peptides from the SP, NP, and MG proteins against HLA-II are shown in (Appendix A).

### 3.6. Most Frequent HLA-1 in LATAM and Recognition of Peptides of Conserved SARS-CoV-2 Regions

The peptide with the highest promiscuity in HLA-I molecules was MPYFFTLLL (n = 66), which was found in the ORF1 transcript obtained from the SARS-CoV-2 proteome. Other peptides included FAMQMAYRF (n = 58), found in SP, and FLLNKEMYL (n = 42), which was located in the ORF1 transcript.

The alleles with the highest promiscuity for all the conserved proteins in SARS-CoV-2 proteomes (restricted to 9 mers) were preferentially found in the C locus, with a Me of 82 agglutinations and 18.2 σ. The molecules were HLA-C*08:03 (n = 98) with the highest agglutination, and HLA-C*04:01 (n = 42) with the lowest agglutination. Locus A follows with a Me of 47 agglutinations and 10 σ. With the molecules HLA-A*34:02, HLA-A*68:02 (n = 62), and HLA-A*36:01 (n = 22) with the highest and lowest agglutination, respectively.

Finally, the B locus with an Me of 40 agglutinations and 15.5 σ. With the molecules HLA-B*35:17, HLA-B*35:20, HLA-B*35:30 (n = 73), and HLA-B*27:05 (n = 8) with the highest agglutination and lowest agglutination, respectively.

In (Appendix A), the total number of antigenic alleles and peptides with a strong binding affinity for HLA-I peptides obtained from the entire SARS proteome-CoV-2 are shown. For a more in-depth analysis of the HLA-I predictions, the alleles are classified from highest to lowest according to their theoretical ability to recognize HLA-I peptides. The proportions refer to allele frequency and locus found in LATAM countries (Appendix A).

For Locus A, the BRA-CHL-COL-CRI-VEN countries have populations that carry HLA-A alleles with a greater ability to recognize HLA-I peptides, while ARG-CUB-ECU-MTQ-NIC-PER has the lower capacity (Appendix A-Locus A). For Locus B, we found that no country had a greater frequency for HLA-B alleles. Rather, LATAM showed a greater ability to recognize HLA-I peptides. The ARG-BRA-CHL-COL-CRI-CUB-NIC-PER countries presented a lower capacity to recognize HLA-B alleles, especially in VEN and CRI. (Appendix A-Locus B).

Regarding Locus C, we had the least amount of data on HLA per country. Nevertheless, it showed a greater consistency, which resulted in a greater ability to recognize HLA-I peptides in BRA-COL-CRI-NIC-PER-TTO. In the case of CHL, HLA-C alleles had a lower capacity to recognize HLA-I peptides. (Appendix A-Locus C).

There could be more susceptibility to mortality and morbidity due to SARS-CoV-2, especially in countries that share more than one locus with less capacity to recognize HLA-I peptides. Such is this case with ARG-CHL-CUB-NIC. The alleles most frequently found in LATAM included HLA-A*02:01, HLA-A*24:02 and, HLA-B*40:02. Based on their capacity to recognize peptides derived from SARS-CoV-2, these alleles were ranked 81st, 119th and, 154th, respectively.

### 3.7. Vaccine Candidates, Post-Translational Modifications, and Experimentally Validated Peptides

Matrices with vaccine potential were constructed to identify peptides associated with the most frequent HLA-I and HLA-II molecules in LATAM. The least number of peptides found to cover the most frequent LATAM alleles is summarized in Table 1.

These PPVPs share characteristics for the construction of multi-epitopes, including being antigenic, non-toxic and, non-allergenic. The first 7 peptides come from structural proteins that are directed towards HLA-II. The remaining 11 come from NSP, except for P9 and P12 that come from SP and ORF7a, respectively, and are directed towards HLA-I. From the SARS-CoV-2 dataset available in IEBD, the “Experimental evidence available” column highlights P10 as recently validated in three patients convalescing from COVID-19 (https://www.iedb.org/assay/12156798, accessed on 3 November 2020). Furthermore, peptides P1, P4, P5, P7, and P9 were harboured in linear epitopes of TLs validated experimentally. P10, P14, and P18 have recently been validated in other HLA restricted studies (https://www.immunitrack.com/free-coronavirus-report-for-download/, accessed on 3 November 2020), Since SARS-CoV-2 can share conserved regions with SARS-CoV and takes longer to study, we investigated whether these PPVPs have been experimentally tested on TLs, HLA, or BLs ligands. With a Blast of 90% in IEBD, peptides P2, P10, and P14 are also conserved in SARS-CoV and have experimental validation. P2 was both HLA and BLs validated. recognized linear epitopes. So far, no specific immunogenic regions or experimental assays have been identified that validate T, B, or HLA cell ligands in the experimental information recovered. On the other hand, P3, P6, and P12 were only partially found in immunogenic regions in experimental SARS-CoV-2 assays. Since they have immunogenic characteristics indicated in the other experimentally validated peptides, these may be PPVPs that could delimit broader regions. Regarding the post-translational modifications, only P5 was found with an N-glycosylation evidenced experimentally in the PDB 6VSB crystallized structure. However, this epitope was completely harboured in a linear epitope in IEBD (test code 8160608), which is capable of being recognized by TLs from convalescent COVID-19 patients. Taking into account that P5 is found in an experimentally validated immunogenic region, it is tentative to think that it is capable of being presented by HLA. As a result of this, it was entered into the construct. In the SP of SARS-CoV-2, we did not identify glycan shields with densities that limit probable antibody recognition [111]. However, its interaction with innate immunity remains unknown. With these modifications, HLA may be able to recognize and present the peptides derived from antigenic processes when they are captured by dendritic cells [112]. These have been most frequently studied in cancer, evidenced in T and BLs primers, which are in some cases even more immunogenic [113,114,115]. Because they are naturally recognized as epitopes, their in-vitro recognition can be included in complementary studies. It quickly identifies the immunogenic regions susceptible to being glycosylated and therefore optimizes the potential vaccine formulations. Finally, in (Appendix A), the HLA-I and II alleles are shown to be strong binders of the PPVPs.

### 3.8. Flexible Peptide-Protein Coupling Signals Favourable Energy and Anchorage Residues to HLA Molecules

To estimate the conformation of the complexes, we used the following allele experimental structures as receptors: DRB1_0401, DRB1_1101, HLA-B*35:01 and HLA-C*07:02 (IDs in RCSB-PDB: 5NI9, 6CPN, 1XH3 and 5VGE, respectively). As peptide ligands, we used SFRLFARTRSMWSFN and FAMQMAYRF.

The best models were those contained in clusters with the highest density, according to the average root-mean-square deviation (RMSD) obtained from each simulation and their adequate position in the HLA cleft. The results are shown in Figure 3.

A map of the contacts between peptides and HLA-I molecules is shown with residues found to more frequently interact in the clusters (functioning as an anchor and generally located in the first and last residues). They also display interactions with β-2 globulin that stabilize the α1 and α2 chains, where the binding groove is formed, and facilitate peptide bonding [67].

In HLA-II, the regions that most frequently interact as anchors are FARTRSMWS for DRB1_0401 and FRLFARTRS for DRB1_1101 (although they differ from the core sequence of the prediction). The residues 4, 5 and 7 most frequently interact, giving a greater number of anchors than those in HLA-I; this results in flanking regions with small deviations in the peptide backbone and differing HLA cleft accommodation.

### 3.9. Molecular Docking, Interactions with Heterodimer TLR-4/MD-2

In the N terminal, β-defensins have a range of immune responses related to cell maturation that mediate innate immunity, such as dendritic cells, TLs, and antiviral activities [116,117]. Another adjuvant used was the universal memory TLs helper peptide TpD, an auxiliary peptide that can aid memory generation as a target of TLs CD4+ [118]. The adjuvant Pan DR T helper epitope PADRE was also attached to the construct. It is relevant in multi-epitope constructs given its ability to potently stimulate the innate and humoral immune systems through high and specific IgG titer generation. It can also overcome barriers, indicated by the high diversity among HLA molecules; hence, it reaches a larger population and is safe [119,120]. Towards the C terminal, a peptide domain, CTGKSC, capable of interacting with M cells and mediating up to an 8-fold increase in intestinal absorption, was added [74]. CTGKSC has been formulated in oral multi-epitope vaccine candidates [74], acting as a transporter towards the M cells of the epithelium associated with the follicle in the Peyer’s patches of the intestine in humans, where it can promote a specific immune response.

The linkers used as spacers between HTL GPGPG and AYY epitopes are suitable and facilitate antigen presentation by directly interacting with transport and assembly mediators to HLA molecules [121]. The di-lysine KK that separates epitopes from BLs is located close to the C terminal. Among the adjuvants used were the EAAAK linkers: efficient separators between the domains present in the multi-epitope construct [122]. The following linkers were used in the vaccine, which contained 510 amino acids: 6 EAAK, 6 GPGPG, 11 AAY, and 5 KK. The proposed construct order is presented in Figure 4.

### 3.10. The Physicochemical Properties of the Multi-Epitope Construct Are Consistent with the Requirements for Generating an Immune Response in an Experimental Model

To generate a safe, stable, and capable construct able to evoke an immune response; the antigenic, non-allergenic, and non-toxic properties of the multi-epitope construct were established using Vaxijen 2.0, Allergen FP 1.0, and ToxinPred. The physicochemical characteristics of the multi-epitope construct, along with special consideration for the final CTGKSC peptide, are shown in Table 2.

In addition to generating a safe construct, it was necessary to identify a thermostable multi-epitope construct, indicated by the aliphatic index for laboratory testing. Solubility and thermostability are associated with adequate overexpression in (*E. coli*) which is the bacteria most commonly used to produce recombinant proteins [123].

Given the parameters from the primary structure of the multi-epitope construct, adequate production in vitro can be inferred because of overexpression in *E. coli* and observed safety in immunological studies.

### 3.11. Three-Dimensional Structure Modeling and Validation of the Multi-Epitope Vaccine Construct

The predicted structural conformation of our construct can be correlated with functional annotation and other multi-epitope constructs. The secondary structure was analyzed using the SOPMA and PSIPRED servers, which revealed the presence of 43% α-helix, 21% β-foil, 30% coils, and 6% β- in the vaccine construction (Appendix A). A reliable approximation of the three-dimensional structure of the construct can be used for further in-depth studies. Molecular dynamics can be used to analyze coupling stability with immunological targets in innate recognition of viral elements as membrane TLR receptors with the production capacity of IFN-1. Therefore, we approximated the tertiary structure using the Robetta server, based on homology models, and we used the Ramachandran diagram and ERRAT to validate the quality of this approach. The results are summarized in (Appendix A).

By using the ClusPro 2.0 server to simulate molecular docking between the vaccine construct and TLR-4, 30 models were generated (Appendix A). These were classified by cluster size according to their representative position. The lowest energy −1192.6 (Appendix A) was found in the sixth cluster with 22 members. The first cluster contained 35 members, which indicates an acceptable probability for the native pose of the complex. The cluster with a balanced adjustment was chosen given that it was closer than adjustments 2 and 3, which were related to a majority of hydrophobic and electrostatic interfaces [124].

This model positions the N terminal of the multi-epitope construct and larger interface area towards the concave side of TLR-4, where its ectodomain forms mostly hydrogen and salt bridges. The free convex side is broken into the N terminal of TLR-4, which indicates the formation of hydrogen and salt bridges resulting from interactions with C-terminal of the multi-epitope construct.

Also, interactions are present in the internal part and run towards MD-2; these are mostly hydrogen and saline bridges. The latter is maintained primarily by the ARG 157 and ARG 159 residues of the SFRLFARTRSMWSFN peptide. These are illustrated in (Appendix A).

### 3.12. Molecular Dynamics

The root mean square deviation RMSD for the vaccine-receptor complex was evaluated during the complete simulation time (Figure 5a). The RMSD plot showed a considerable increase until it stabilized at around 20 ns, with an average of about 0.8 nm for the rest of the simulation time. These changes indicate that the vaccine attempts to find the best position based on its receptor. After 20 ns from the beginning of the simulation, it remained stable, which is an indicator that the vaccine reaches the best conformation to form a stable complex. Also, the root-mean-square-fluctuation (RMSF) of the vaccine and its receptor was evaluated. Residues from the vaccine start at GLY-1483 and finalize at residue CYS-1992 (Figure 5b). The RMSF showed a stable conformation from residue GLY-1483 to LYS1683 from the vaccine segment, which are part of the β-defensin 3 to TpD of the vaccine construct, Figure 4a. This stabilization is made due to the salt bridges formed in the residue ASP-1424:ARG1639, ASP-756:ARG-1641, GLU-1180:ARG-1524, GLU-1183:ARG1520, and GLU-1556:ARG930, which showed regular contact during the entire simulation (Appendix A). On the other hand, a segment of the vaccine, which corresponds to residue PHE-1684 to CYS-1992 (Figure 5b), showed a higher RMSF, displaying a pronounced motion compared with the GLY-1483 to LYS1683 segment, while the TLR-4 receptor remained stable. Apart from this motion, the complex remained stable during the 68 ns of simulation.

The radius of gyration (Rg) was calculated to determine the compactness of the vaccine-receptor complex system during the simulation (Figure 5c). It did not show a significant change. The complex exhibits a compact folded structure during the simulation, with an average Rg of approximately 4.8 nm. This folded complex structure is stabilized on average by around 9 H-bonds (Figure 5d) and 5 salt bridges, which were described above. Thus, the Vaccine-TLR4 complex showed a stable conformation during the 68 ns of simulation.

### 3.13. Optimization of cDNA from the Vaccine Construct for Optimal Expression of the Vaccine Product

For insertion of the vaccine construct into a plasmid vector, the CTGKSC+ protein sequence of 510 amino acids was inversely translated to a cDNA of 1530 nucleotides in length. The host system of expression varies, and the cDNA must adapt according to the use of the host codon. For optimal expression of the multi-epitope construct in the *E. coli* K 12 host, the resulting cDNA was codon-optimized according to the JCAT server. Furthermore, optimization of the rho-independent transcription terminator and prokaryotic ribosomal binding sites in the middle of the cDNA sequence were avoided to generate optimal and complete protein expression. To insert the construct into the pET28a(+) cloning vector, the BamHI and HindIII cleavage sites were also avoided. The results are shown in (Appendix A).

### 3.14. Expression of the Multi-Epitope Vaccine Construct in E. coli K 12 by In Silico Cloning

The *E. coli* K 12 strain was selected as the cloning organism since multiple-epitope vaccines are expressed and purified more easily in this bacterium. For this purpose, the expression vector pET28a(+) was used and excised with the restriction enzymes BamHI and HindIII. The optimized cDNA was then inserted near the ribosome binding site using Snapgene (Appendix A).

### 3.15. Immune System Simulation

To identify the immunogenic profile of the vaccine construct, we used the C-IMMSIM immune server. As shown in Figure 6, the secondary and tertiary responses showed greater global responses and the presence of memory T and BLs. This may have been due to the cumulative effect of cells at the serum level that possesses a memory profile exceeding the total in injection 3. This is also associated with decreased antigenic concentrations and normal immunoglobulin levels. Cell-specific lineages were also stimulated, with the general activation of CD4+ and CD8+ TLs. Also, HTL-1 cell-based immunity was predominant. This is associated with the production of the cytokines and interleukins IFN-1, TNF-β and IL-2, as well as the activation of professional antigen-presenting cells.

These results are consistent with the construct having a natural immunogenic capacity, which was applied without Lipopolysaccharide (LPS) in the simulation. A more extensive immune simulation corroborates the memory formation indicated in this brief approach. It was conducted until day 311 with 12 injections, in a step of time beyond 460 days, adjusting the intervals that did not surpass 4 weeks, 12, 94, 178, 262, 346, 430, 514, 598, 682, 766, 850, 934 (Appendix A).

## 4. Discussion

In this study, we used a reverse engineering approach to design a multi-epitope vaccine for SARS-CoV-2 based on the identification of PPVPs (Table 2) from the conserved regions of proteins in 92 SARS-CoV-2 proteomes. Besides, we added potential epitopes for BLs, adjuvants, and linkers seeking a specific immune response (Figure 5). By using predictors with a higher experimental correlation against HLA-I, we can describe possible considerations in LATAM populations according to the locus and most frequent alleles found (Appendix A). Our approach differs from other in silico vaccine models proposed for SARS-CoV-2 because we considered the total of structural, non-structural, and accessory proteins to identify potential vaccine candidates. This not only aims to neutralize the input [125] or most immunogenic proteins [126], but also those related to replication and immune modulation. This offers broader protection without making a large protein since we used the least number of peptides capable of covering all HLA alleles most frequently found in 18 LATAM countries. In Table 1, P1 to P7, which targets HTL, aims to stimulate LT and activate BL. P8 to P18 are PPVPs directed towards CTL that result from the cleavage of polyprotein 1a/b. P8 and P18 from NSP did not result in luminal subcellular locations resistant to post-translational modifications. In addition, P10, P14, and P18 obtained through the similar algorithms used in this study were already experimentally validated.

Other authors have previously identified peptides from SARS-CoV-2 using similar bioinformatic tools similar to ours and have classified possible vaccine targets such as P9, P7, P5, P18 (in SP), and P12 (in ORF7) [127,128,129,130,131,132]. The multi-epitope construct followed by PPVPs in the linear epitopes of SP and NP with the capability of interacting directly with the LB receptor is presented in (Appendix A). In SP, linear epitopes were taken by consensus, occupying the RBD and lacking N-glycosylations, thereby attempting to disrupt ACE2 coupling by stimulating specific antibodies with neutralizing capacity. However, it has been identified that specific antibodies against RBD are not the only way to neutralize SARS-CoV-2 [133]. The experimental evidence available in the IEBD column of Table 2 indicates peptides with experimental validation in both SARS-CoV and SARS-CoV-2, far from RBD such as P2. This was complemented with experimental tests on BLs and HLA. Another peptide, P9, shares similar characteristics but is only partially conserved, delimiting only a potential immune region. These indicate that other regions are capable of stimulating LB, LT, and HLA shared by SARS-CoV-2 and SARS-CoV, far away from RBD, which means that P2 could generate antibodies towards SP outside of RBD. Recently, 61 monoclonal antibodies isolated in serum from patients with severe COVID-19 requiring mechanical ventilation have been identified; 9 antibodies exhibited a high neutralizing capacity against SARS-Cov-2 in vitro, 5 were directed at RBD, 3 against the N-terminal domain, and 2 against indeterminate regions of [133]. Other authors have confirmed the neutralizing capacity of directed antibodies towards the N-terminal when isolating it from convalescent COVID-19 patients [134]. Our study consists entirely of bioinformatic analysis, but some PPVPs resulted from experimental evidence. Some PPVPs that were not found in the experimental databases consulted were P3, P6, P8, P11, P12, P13, P15, P16, and P17. It is important to gather information on these PPVPs as potential vaccine targets because they share common immunogenic characteristics.

HTL with antigenic properties primarily comes from SP (n = 114), MG (n = 62), and NP (n = 23). In comparison, the results for CTL were ORF1a/b (n = 573), followed by SP (n = 63), and NP (n = 34). Most of these epitopes originated in the ORF1a/b transcript or from MG and SP. Compared to other proteins like accessories or other structural proteins, these could harbour a greater number of conserved immunogenic regions because they were found in predictions with greater agglutinations against LATAM alleles.

In this study, we corroborated NP as a relevant vaccine target. We found areas in NP with predictions of linear and discontinuous epitopes longer than those of SP, resulting in a greater chance of antibody production. We also highlighted the immunogenicity of SP and identified a greater number of strong agglutination predictions in association with more frequent HLA-II molecules in LATAM. We emphasize that the majority were found outside the receptor-binding domain. In SP, an interval between residues 492 and 524 was found to contain several PPVPs associated with HLA-I and II molecules. This was also found in the most frequent phenotypes in LATAM, e.g., HLA-B*16:02. This is an important immunogenic region that is also associated with the linear epitopes of BLs. However, it is important to keep in mind that using the complete SP subunit could be more effective in a vaccine formulation. It should be noted that mutations in this immunodominant region could imply changes in antigenic recognition with loss of protective capacity [135]. Our multi-epitope construct offers a specific immune response with greater attention to stimulating innate immunity, especially against relevant proteins in the phase of infection. Reflecting on whether it can offer cellular protection, even if antigenic drift or natural selection or by any SARS-CoV-2 vaccine itself throughout the pandemic or after, it may exceed the protection offered by neutralizing antibodies against the RBD. Protection that is not expected at least directly, when using the recombinant SP protein present in current commercial formulations obtained from the Wuhan-Hu-1 sequence [136]. Multi-epitope constructs, including the one proposed here, if required, can be periodically updated specifically covering new conserved B or TLs epitopes that emerge in SARS-CoV-2 variants, avoiding a delayed elimination of infected cells. Since by not using all the SP protein, other epitopes of SP present in most variants can be raised, without falling into the effect of the original antigenic sin [136,137,138]. Although we studied the most frequent alleles in LATAM, we only have one SARS-CoV-2 proteome from Brazil in the sample of proteomes (this can be considered as a drawback of the study, because of the limited amount of data to date), that were selected for the in silico analysis described in the pipeline (Figure 1—Pipeline) and available in (Appendix A). We used a conservative approach by eliminating the non-synonymous amino acid substitutions presented in the 92 proteomes collected between December 2019 and 11 March 2020, from other regions of the world, mainly Asia. These blocks of conserved non-redundant amino acids were used for immunoinformatics analysis to avoid reformulations of the multi-epitope construct. This is due to the uncertainty about certain variants that could become more frequent over time and occupy immunogenic regions of SARS-CoV-2 throughout the ongoing pandemic. According to GISAID statistics [139], a study in the US by Wang et al. [140], and in Uruguay by Elizondo et al. [141], for the last week of October 2020, there was no indication of an aa-changing SNP with a clear trend in PPVPs and for LB epitopes (Appendix A) in the vaccine construct. Although the US and Uruguay had introductions of SARS-CoV-2 from different geographical areas of the old world [141,142], there is evidence of similarity between the aa-changing SNP with the global ranking exposed by GISAID. The approach used remains current and utilizes data from around the world despite coming from analysis with a reduced number of proteomes in an early stage of the pandemic [139]. As well as being the result of the analysis of a group of the most frequent HLA alleles in LATAM. We added catalogued safe adjuvants to increase immunogenicity and overcome the HLA polymorphic barrier, which is typically an obstacle to the development of epitope-based vaccines [143]. Some HLAs and supertypes have been associated with a lack of response, e.g., the oral rotavirus vaccine in infants [144]. PADRE has been shown to improve the immune response in human papillomavirus vaccines by generating a robust CD8+ response and in hepatitis B virus vaccines by improving the presentation of epitopes and thereby generating specific CTLs [145,146]. Human β-defensin 3, which activates human monocytes dependent on TLR1/2 [147], is considered an endogenous adjuvant as it induces the maturation of Langerhans cells to dendritic cells and stimulates these cells to induce strong proliferation and IFN-γ production by CD4+ TLs [148]. This approach was previously used to construct bovine herpes 1 DNA vaccine, in which it increased the production of IFN-γ-dependent LT CD8+ [149]. We also used TpD as an adjuvant, which has previously been studied extensively in animal models and peripheral blood samples. This resulted in increased TLs and the robust generation of antibodies [118]. CTGKSC theoretically increases the water solubility of the vaccine proposal (Table 2) due to its amphiphilic properties [74] and it could better deliver the multi-epitope construct to TLR-4/MD-2 in the M cells of the Peyer’s patches follicles using polymeric [74,150,151]. With this alternative delivery and validation of the vaccine proposal, a greater reach could be achieved to include remote populations with difficult access to medical assistance.

The final construct was catalogued as antigenic, non-allergenic, and non-toxic, which establishes it as a safe and powerful vaccine candidate against SARS-CoV-2 (Table 1). Furthermore, the adjuvants provided in the multiepitope construct allows for a synergistic effect with other types of adjuvants. Proposals for multi-epitope peptide vaccines similar to that of this study have been proposed for various infectious etiologies (DEN-2 [152], Hendra [153], Nipah [154], *Pseudomonas aeruginosa* [77], *Klebsiella pneumoniae* [155], *Plasmodium* spp. [156], and non-infectious (Kaposi’s sarcoma [157]). For in vivo models, multi-epitope vaccines for influenza virus type A [158], Ebola virus [159], HPV-16 [160], and Uropathogenic *E. coli* [161] have been tested and have shown to be capable of stimulating the innate and adaptive immunity to ensure specificity and safety.

In SARS-CoV-2, the validation of immunogenic peptides obtained from structural, non-structural, and accessory proteins was carried out. This was based on a bioinformatic analysis, similar to ours, which demonstrated the activation of TL in the peripheral blood of convalescent Covid-19 patients at the University Hospital Tuebingen [162]. From these findings, the authors selected 8 peptides capable of interacting and activating TL-CD4, TL-CD8, and producing IFN-γ. These were incorporated together with the XS15 adjuvant, emulsified in MONTANIDE^TM^ ISA 51 VG, which targets TLR1/2. This approach is currently in Phase 1 trials (NCT04546841) [163]. Another study in the preclinical phase focused on optimizing a vaccine for TL-CD8 using artificial intelligence and computational algorithms like the aforementioned study. The authors also used conserved regions of structural, nonstructural, and accessory proteins for the binding prediction to the HLA-A*02:01 allele, one of the most common worldwide. Using an in vivo methodology, the authors utilized an HLA-A*02:01 transgenic mouse, which indicated an activation and production of memory TL to 55 peptides coupled to the PADRE sequence together with the Montaigne adjuvant. In patients with a history of symptomatic, moderate, and severe COVID-19, there was at least one and a maximum of six months without evidence of infection. These patients were shown to activate peripheral blood monocytes towards these peptides [164].

Using an immune simulation with a heterozygous HLA adjustment, represented by some of the most frequently recovered LATAM alleles, we confirmed the immunogenic nature of the vaccine constructs. Without the extra adjuvant LPS, the vaccine constructs promoted an immune profile that indicated adequate antigenic presentation and the subsequent generation of immune memory in groups of T and BLs. Also, there was a polarized response towards HLT-1 and the stimulation of several immunoglobulins (IgG1+IgG2, IgM and, IgG+IgM) following the first injection. There was a robust response to the third injection, with evidence of the production of IFN-1 and IL-2 cytokines related to CD8+ lineage expansion. Comparative simulations using the C-IMMSIM algorithm have been conducted with experimentally validated peptides. Correlations with in vitro studies were found [165], which suggests that in vitro experiments with our construct may potentially evoke similar cellular behaviours.

These predictions are consistent with the analysis of flexible protein-peptide docking, which provides evidence of the adequate anchorage of the side chains of some residues of two promiscuous peptides located in the vicinity of the single groove of different HLAs. This results in adequate accommodation of the core sequence in HLA-I and II molecules. It is associated with effective immune interaction, and therefore sufficient presentation of CD4+ and CD8+ TLs, that when is activated, trigger an immune response [67].

Our in-depth secondary structure analysis showed the predominance of α-helices, while the tertiary structure showed an optimal spatial arrangement of amino acids. The modelled structure was further improved which increased its general quality. When we performed a molecular coupling of TLR-4/MD-2 using ClusPro v2, the resultant interfaces were generally based on hydrogen and salt bridges supported primarily by TLR4. This fixes the multi-epitope construct towards the concave and terminal C region of its ectodomain, identifying a probable non-canonical interaction. At this location, some P6 residues of MG have more contact near the more hydrophobic region of MD-2, where coupling with LPS has been reported [166]. Therefore, this interaction could be stable and safe since it shows the hydrophobic pocket of empty MD-2 which is known to interact with traces of LPS and is associated with a fatal toxic immune response [167]. Other protein interactions that occupy TLR-4 could be safety-relevant. In COVID-19 an immunopathological process has been described that at least theoretically could be explained in part from the interaction of SP with TLR-4, increasing the production of inflammatory cytokines secondary to the formation of interfaces composed of antigenic residues of SP, interacting with external residues of TLR-4 [168]. In our multi-epitope construct, although we use antigenic peptides that come from SP, none of these antigenic residues is part of the interfaces described.

In nature, other non-canonical interactions have been described as in the case of the HIV Tat protein that results in an interface between the N-terminal and TLR-4/MD-2, showing greater stability in the resulting complex [169]. Also, other studies on larger and smaller viral proteins, including Tat [170,171], have identified an inhibitory competition of the binding site towards MD-2 by lipid A of LPS from *Rhodobacter sphaeroides*. This could affect the activation by disrupting the formation of a stable complex with TLR-4. However, the MD-2 interaction may not always be necessary for receptor activation. For example, in *E. coli*, the adhesive subunit FimH can evoke a more immunogenic profile against LPS by targeting TLR-4, which up-regulates the expression of HLA-I and II molecules [172]. Therefore, the stable and low entropy formation between the interfaces of the tetramer with potential ligands, including multi-epitope constructs, could activate it [173,174]. The position and situation that our multi-epitope construct occupies has been found in other vaccine models constructed using different methods. Therefore, it seems to be relevant to the stabilization of the multi-epitope construct [165]. In addition to the immunogenicity and stability identified in our molecular dynamics analysis, the physicochemical analysis of the construct revealed other desirable characteristics related to a safer profile. In particular, the construct was found to be nontoxic and nonallergenic. It was also thermally stable, which suggests that it would be suitable for the overexpression in the bacterial model *E. coli* K 12. A successful in silico cloning procedure is a cost-effective option that can be extrapolated to laboratories in LATAM countries. Based on the algorithms used in this study, some considerations regarding susceptibility and the expected clinical result should be used discreetly. We identified the HLA-C Locus as the highest loading capacity of antigenic peptides that are predicted to strongly binding towards the most frequent HLA-I alleles in LATAM from conserved regions of SARS-CoV-2 (Appendix A). However, the HLA-C promoter lacks binding sites to the transcription factor NF-κ
β [175], which is upregulated in the antiviral state. Therefore, it may not represent an accurate picture of an active infection. Although the C locus can present antigenic peptides and activate CTL, this locus is more in tune with the effector function of natural killers [176], which could regain importance in the activation of NK in COVID-19. However, a greater presence of antigenic peptides is necessary for its membrane expression, and its high capacity to recognize peptides from SARS-CoV-2 could be associated with less thymic selection. Interestingly, the HLA-B*35:01 allele, which was found in one of the HLA molecules capable of recognizing a greater number of peptides from SARS-CoV-2, was found more frequently in BRA-CHL-COL-CUB -NIC-PER (Appendix A). It was recently recognized in a cohort of Spanish patients with moderate COVID-19, which did not have a fatal outcome [107]. According to the country of origin, when compared to Locus C and A, locus B did not represent a group of alleles in LATAM with a higher carrying capacity of peptides. As a result, despite the country of origin, certain people will have a better outcome from SARS-CoV-2. It would be interesting to review BRA-CHL-COL-CRI-VEN in the future. Especially VEN, since it was found with alleles belonging to Locus A, which can recognize many peptides from SARS-CoV-2. Countries such as ARG-CUB-ECU-NIC have a lower capacity to recognize alleles from locus A. Additional attributes of HLA molecules found in HLA-B*35:01 include stability in acidic pH and alternate antigenic presentation [177] and allelic heterozygosity in patients with COVID-19, which could bring us closer to a precise treatment. This type of approach can better characterize populations at risk and allow a deeper understanding of the immunopathology caused by SARS-CoV-2. A collaborative initiative on the study of HLA and COVID-19 is currently available at (http://www.hlacovid19.org, accessed on 19 December 2020).

## 5. Conclusions

SARS-CoV-2 undoubtedly poses a substantial social challenge for all citizens of the world. Not only are its effects devastating to public health, but they have also set back the progress made over decades in other fields like economics, equity regarding social determinants in health and social management of other endemics outbreaks, especially in regions with emerging economies. By using bioinformatics tools and considering the immunological profiles of vulnerable and diverse regions in LATAM, the use of peptide binding predictors such as SARS-CoV-2 and HLA-I can be considered. This can be used as part of a precise treatment plan focused on characterizing people and populations with a greater risk of COVID-19 infection.

A vaccine is the best option for limiting the medium- and long-term effects of COVID-19 on the global population. We created a candidate multi-epitope vaccine directed towards the world population, with an immunogenic capacity that surpassed the safety standards required to begin complementary experimental studies. Furthermore, we try to analyze the binding affinity of potential antigenic peptides resulting from our pipeline of most prevalent HLA molecules in LATAM as part of a regional immunoinformatics approach. Several PPVPs that resulted from this methodology were found using experimental validation and demonstrating others that share similar immunogenic characteristics and should be validated.

We hope to quickly validate the vaccine candidate and take action against the ongoing pandemic.

## Figures and Tables

**Figure 1 vaccines-09-00581-f001:**
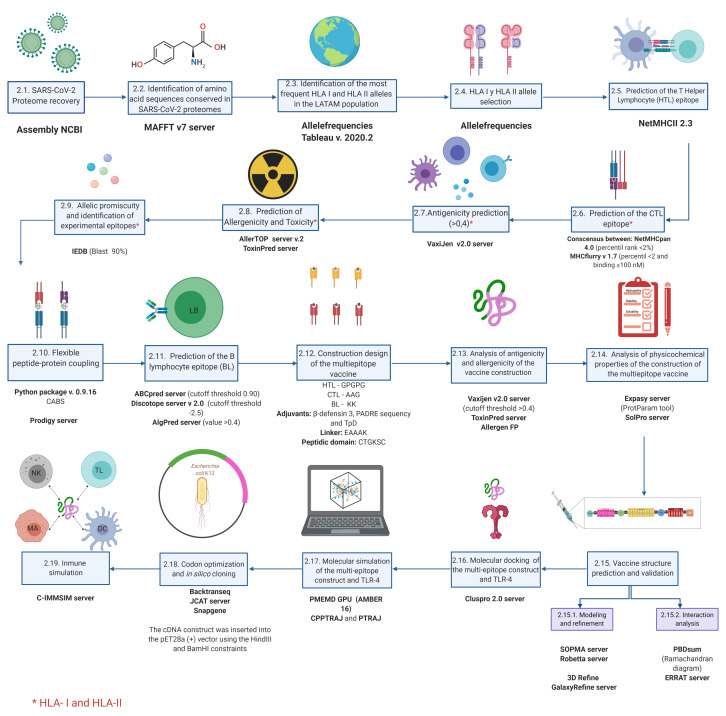
Vaccine development pipeline: the design stages of the multi-epitope vaccine proposal are shown. Each stage is important for meeting the objective of identifying peptides capable of generating an immune response, taking into account the most frequent alleles in Latin America. The important servers and cutoff values used in the process are specified.

**Figure 2 vaccines-09-00581-f002:**
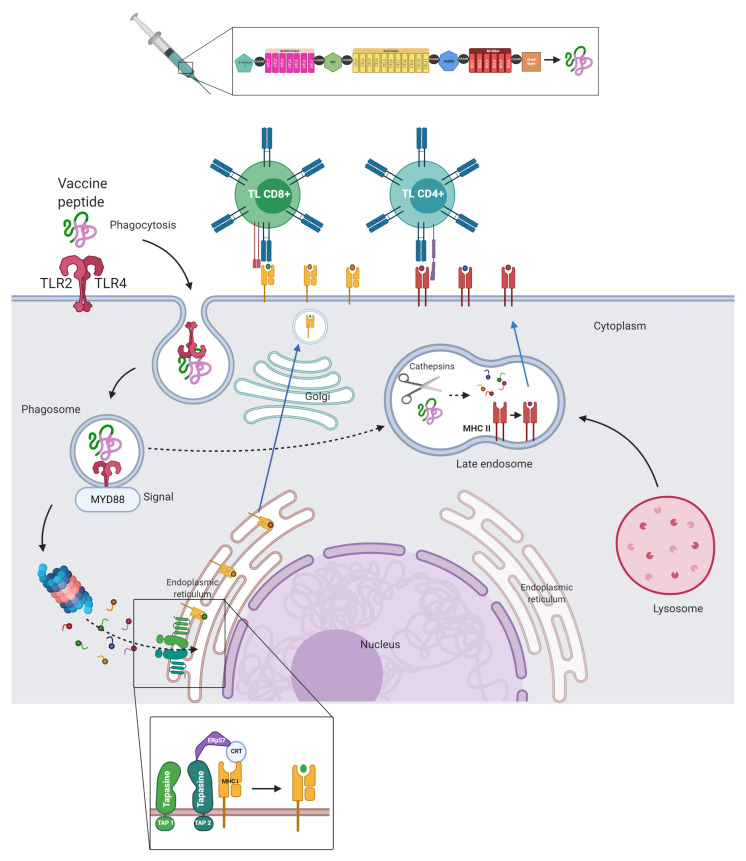
Antigenic presentation from the proposed vaccine: The assembly of peptides with vaccine potential are recognised by cell membrane receptors capable of recognising patterns associated with pathogens, such as TLR2 and 4 from dendritic cells. After recognition, the construct is phagocytized by the cell together with TLR, allowing its interaction with MyD88 and the maturation of the phagosome. From the phagosome, the peptide can take two routes: the first route is towards the proteasome, where the peptide is degraded, internalised in the ER and assembled with HLA-I molecules; the second route involves the internalization of the peptide in the late endosome, where it is assembled with HLA-II molecules and subsequently presented on the cell membrane of the LT [103,104,105].

**Figure 3 vaccines-09-00581-f003:**
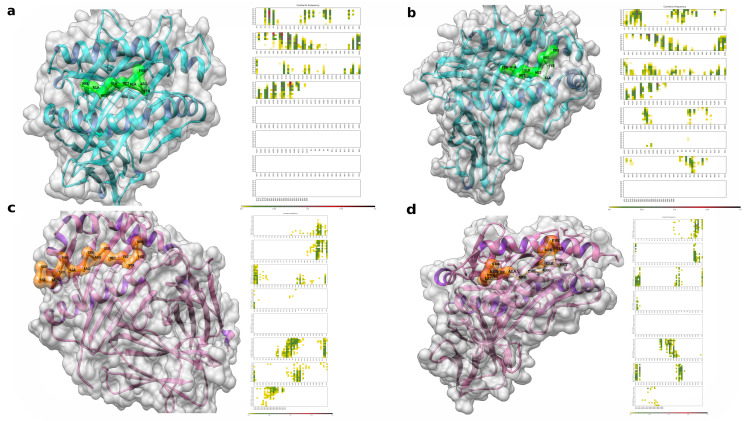
Estimation of flexible molecular protein-peptide coupling: In the complexes (upper panels) are two non-redundant HLA-I molecules most frequently found in the Latin American population coupled with a peptide with vaccine potential from SP, namely FAMQMAYRF. (**a**), Cluster 1, density of 239, ΔG of −5.2. (**b**), Cluster 1, density of 355, ΔG of −4.9. In the complexes (lower panels) are two non-redundant HLA-II molecules most frequently found in the Latin American population coupled with a peptide with vaccine potential from MG, namely SFRLFARTRSMWSFN. (**c**), Cluster 1, density of 138, ΔG of −4.6. (**d**), Cluster 2, density of 168, ΔG of −4.4.

**Figure 4 vaccines-09-00581-f004:**
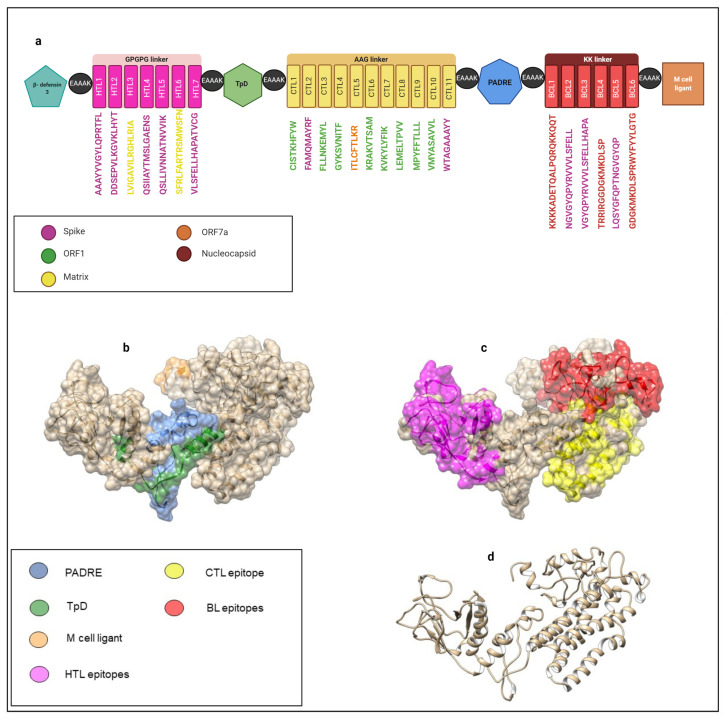
Construction of the Proposed Multi-Epitope Vaccine. Structure of the multi-epitope construct: three-dimensional diagrammatic structure of the multi-epitope construct showing (**a**), the separators, origin of the peptides and adjuvants used; (**b**), the location of the adjuvants PADRE, TpD and M cell ligand; and (**c**), the location of the epitope’s cytotoxic T lymphocytes (CTLs), T helper lymphocytes (HTLs) and B lymphocytes (BLs). (**d**), Ribbon diagram of the multiepitope construct.

**Figure 5 vaccines-09-00581-f005:**
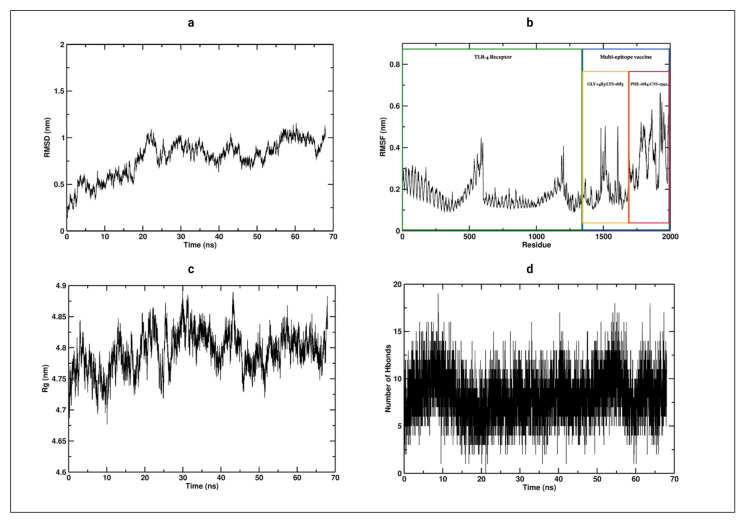
Analysis of trajectory. (**a**), Root mean square deviation (RMSD) Cα atoms. (**b**), Root mean square fluctuation for Cα atoms (RMSF), TLR-4 receptor initialize at amino acid 1 and goes to amino acid number 1482, and the vaccine from amino acid number 1483 to 1992. (**c**), Rg plot; vaccine construct is stable in its compact form during the simulation time. (**d**), Changes in the number of hydrogen bonds between the TLR-4 receptor and multi-epitope vaccine molecule during MD simulation.

**Figure 6 vaccines-09-00581-f006:**
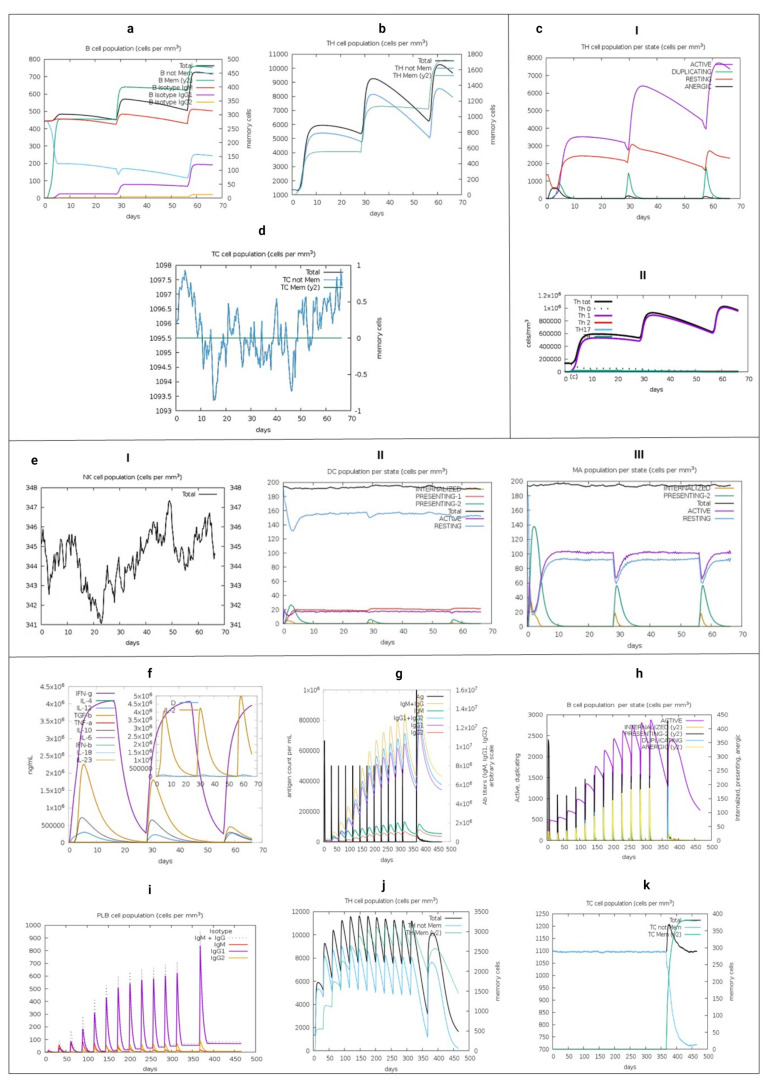
Immune simulation using the multi-epitope construct. (**a**), BLs, (**b**), CD4+ TLs and (**d**), CD8+ TLs were simulated, presenting a cumulative effect towards the third injection on the 56th day of simulation. This suggests the early presence of TLs memory and a change towards the immunoglobulin isotype, immunoglobulin G (IgG), being more predominant. While the antigenic stimulus lasts, there is a polarization towards a certain type of response (**c**), HTL-1, which is consistent with (**e**), the active antigenic presentation of professional antigen-presenting cells. This could be partly stimulated by other non-presenters, as well as the production of interleukins, such as (**f**), IFN γ, TGF-β and IL-2. An infection challenge, composed of a virus responding to the sequence of SARS-CoV-2 proteins covered by the multi-epitope construct, was simulated on day 366. (**g**), An indifference in immunoglobulin M (IgM) production and increased IgG response suggests favourable conditions for viral antibody clearance in the BLs. (**h**), The duplication and antigenic presentation of BLs correspond to the stimulated response by the simulated virus, which lasts beyond the viral challenge. (**i**), An IgG isotype shows a substantial response to the viral challenge, with subtype IgG1 most notably responding. (**j**), The CD4+ TLs population is globally stimulated; the memory response results in consolidation, which increases cell numbers and is still available over 100 days after the viral challenge. (**k**), The population of memory CD8+ cells is stimulated by the viral challenge, acting directly on viral clearance.

**Table 1 vaccines-09-00581-t001:** Potential promiscuous vaccine peptides for the most frequent HLA-I and HLA-II alleles in LATAM. Position: Refers to the first and last amino acid residue in the range occupied by the peptide in the source protein. Source protein: Refers to the proteins identified from the proteomes obtained from NCBI. Vaxijen: Antigenicity described by a threshold greater than 0.4. AllerTOP: Prediction of whether the peptide is a probable allergen. HLA: Experimental database referenced in methodology as Immunitrack, in which the stable coupling of the peptide with at least one HLA molecule is evidenced. IEBD: Refers to the identification of peptides at the base of IEBD specifying if the peptide is found as a ligand of HLA, BL, or TL or if it is contained in an immunogenic region validated experimentally by any assay. * Peptide is conserved in SARS-CoV-2 and SARS-CoV.

Number of Peptide	Peptide	Positions	Protein of Origin	Vaxijen	AllerTOP	HLA	IEBD
1	AAAYYVGYLQPRTFL	262–276	SP	0.48	Non-Allergen	n/a	Validated region
2	DDSEPVLKGVKLHYT *	1259–1273	SP	1.18	Non-Allergen	n/a	Validated peptide—Assay in BLs
3	LVIGAVILRGHLRIA	138–152	MG	0.88	Non-Allergen	n/a	n/a
4	QSIIAYTMSLGAENS	690–704	SP	0.57	Non-Allergen	n/a	Validated region
5	QSLLIVNNATNVVIK	115–129	SP	0.43	Non-Allergen	n/a	Validated region
6	SFRLFARTRSMWSFN	99–113	MG	0.80	Non-Allergen	n/a	n/a
7	VLSFELLHAPATVCG	512–526	SP	0.48	Non-Allergen	n/a	Validated region
8	CISTKHFYW	3147–3155	ORF1-NSP4	1.90	Non-Allergen	n/a	n/a
9	FAMQMAYRF	898–906	SP	1.03	Non-Allergen	n/a	Validated region
10	FLLNKEMYL *	3183–3191	ORF1-NSP4	0.44	Non-Allergen	HLA-A*02:01	Validated peptide—Assay in TLs.
11	GYKSVNITF	835–843	ORF1-NSP3	2.37	Non-Allergen	n/a	n/a
12	ITLCFTLKR	110–118	ORF7a	2.02	Non-Allergen	n/a	n/a
13	KRAKVTSAM	4022–4030	ORF1-NSP8	0.76	Non-Allergen	n/a	n/a
14	KVKYLYFIK *	4225–4233	ORF1-NSP9	1.06	Non-Allergen	HLA-A*11:01	Validated peptide—Assay in TLs.
15	LEMELTPVV	1012–1020	ORF1-NSP3	1.97	Non-Allergen	n/a	n/a
16	MPYFFTLLL	2169–2177	ORF1-NSP3	0.49	Non-Allergen	n/a	n/a
17	VMYASAVVL	3684–3692	ORF1-NSP6	0.48	Non-Allergen	n/a	n/a
18	WTAGAAAYY	258–266	SP	0.63	Non-Allergen	HLA-A*01:01	Validated region

**Table 2 vaccines-09-00581-t002:** Physicochemical characteristics of the multi-epitope construct. The default thresholds for ToxinPred, Vaxijen > 0.4, and allergen FP were chosen to be the best balance between specificity, sensitivity, and precision while taking account of linear and non-linear reasons in the results. Besides, the physicochemical characteristics, construct, antigenic, non-allergenic, non-toxic, soluble, and stable characteristics are demonstrated suggesting that interaction with the immune system can allow and maintain recognition by the innate immune system, as well as by BL.

Peptide C Terminal	CTGKSC+	CTGKSC−
Vaxijen V2.0	0.6419	0.6455
AllerToP V2.0	Non-Allergenic	Non-Allergenic
Allergen FP V1.0	Non-Allergenic	Non-Allergenic
ToxinPred	Non-Toxic	Non-Toxic
	Physicochemical Parameters	
GRAVY	−0.038	−0.033
Molecular Weight	55.67 kDa	54.62 kDa
Stability	33.76 (<40)	33.47 (<40)
Solubility	72%	65%
Half-Life (reticulocytes)	30 h	30 h
Aliphatic index	81.81	82.62
Size (amino acids)	510	499

## Data Availability

The proteomes of 92 SARS-CoV-2 strains were downloaded from the Assembly database available at the National Center for Biotechnology Information (NCBI) (https://www.ncbi.nlm.nih.gov/assembly, accessed on 27 April 2020).

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
