# Peer review of "An Immunoinformatics Approach for SARS-CoV-2 in Latam Populations and Multi-Epitope Vaccine Candidate Directed towards the World’s Population"

_vaccines, 2021, doi:10.3390/vaccines9060581_

Round 1

Reviewer 1 Report

This is an interesting manuscript that proposes a rational multi-epitope candidate vaccine against SARS-CoV-2 using bioinformatics. The research was conducted in LATAM, studying the most common HLA I and II in the population to predict immunological complexes among peptides extracted from the conserved regions of 92 SARS-CoV-2 proteomes. The aim was to propose a vaccine using the least number of peptides capable of covering HLA alleles most frequently found in 18 LATAM countries.

Here I have some concerns

The manuscript is good and original, however it can be improved by narrowing the focus. It is too long . The paper covers a lot of topics, from alleles frequency to detailed explanations about immune response pathways. This result in a great amount of inputs that divert reader’s attention from the focus of the research.

-           

:

LINES 84-87

In the “RECOVERY” study, oral or intravenous dexamethasone 6 mg/day was associated with a reduction in the mortality of patients requiring assisted ventilation and who were in the second week of clinical diagnosis; both the conditions need to be satisfied. I suggest to change “as well as” with “and” not to lead to misunderstandings.

For more exhaustiveness, here is the original article: DOI: 10.1056/NEJMoa2021436

LINES 96-100

Also a recent Italian study “TSUNAMI” did not show a plasma benefit in terms of reducing the risk of respiratory worsening or death in the first thirty days.

Source: https://www.aifa.gov.it/documents/20142/1289678/Comunicato_AIFA_641_EN.pdf

LINES 100-109

When speaking about “off label” approaches being evaluated worldwide, besides Tocilizumab and mesenchymal cells, also other treatments deserve to be mentioned. This article has an exhaustive table with current COVID-19 drugs under clinical investigation. DOI: 10.3892/ijo.2020.5159

LINES 123-124

The author states that NP develops a memory response from TLs that remain up to 11 years after infection. I think this sentence needs to be better clarified. After SARS-CoV infection (not SARS-CoV-2) it has been demonstrate that all memory T cells detected were specific against SARS-CoV structural S, N and M proteins (not only NP). Here is the original article for a better understanding: DOI: 10.1016/j.vaccine.2016.02.063

LINE 136

Typing error: change “o” with “or”.

LINES 136-137

Why using NP as a vaccine target for SARS-CoV-2 is considered less safe when compared to SP? I think the author should cite a source, here.

LINE 136

Typing error: delete “however,”.

LINES 435-444

I think the fact that – of the 92 isolates of SARS-CoV-2 - only one came from LATAM, should be mentioned as a limitation of this study.

Author Response

We are very grateful for the comments made on the manuscript, here you will find the answers to your concerns:

Corrections are made in blue color, as was changed in the main document

This is an interesting manuscript that proposes a rational multi-epitope candidate vaccine against SARS-CoV-2 using bioinformatics. The research was conducted in LATAM, studying the most common HLA I and II in the population to predict immunological complexes among peptides extracted from the conserved regions of 92 SARS-CoV-2 proteomes. The aim was to propose a vaccine using the least number of peptides capable of covering HLA alleles most frequently found in 18 LATAM countries. Here I have some concerns
The manuscript is good and original, however it can be improved by narrowing the focus. It is too long . The paper covers a lot of topics, from alleles frequency to detailed explanations about immune response pathways. This result in a great amount of inputs that divert reader’s attention from the focus of the research.

LINES 84-87

In the “RECOVERY” study, oral or intravenous dexamethasone 6 mg/day was associated with a reduction in the mortality of patients requiring assisted ventilation and who were in the second week of clinical diagnosis; both the conditions need to be satisfied. I suggest to change “as well as” with “and” not to lead to misunderstandings. For more exhaustiveness, here is the original article: DOI: 10.1056/NEJMoa2021436

Currently, only Dexamethasone and Remdesivir have been shown to modify the 
84 natural course of the disease. The “RECOVERY” randomized clinical trial (RCT) was 
85 carried out in England. In this study, oral or intravenous Dexamethasone 6 mg/day was 
86 associated with a reduction in the mortality of patients requiring assisted ventilation, 
87 and patients who were in the second week of clinical diagnosis [24]. Another 
88 study, “ACTT1”, involved 10 countries in a multicenter RCT. The trial showed that the 
89 nucleotide analog Remdesivir, which used a loading dose of 200 mg and a maintenance 
90 dose of 100 mg/day, significantly reduced the time it took for patients to recover from 
91 COVID-19. Additionally, the need for supplemental oxygen after 10 days of intravenous 
92 treatment diminished [25]. However, the results of the WHO multicenter RCT “SOLIDARITY” study on the evaluation of the effect of “off label” drugs, which included the follow-up of about 11,330 adults from 4 continents, 30 countries and 405 hospitals, did not find Remdesivir to be associated with substantial prevention of in-hospital mortality, benefiting only a small fraction of patients when using a loading dose and standard maintenance \cite{who2021repurposed}.
92 Currently, the Food and Drug Association has approved 
93 the use of Remdesivir and Dexamethasone for COVID-19, although the latter is only a 
94 temporary approval

LINES 96-100 and LINES 100-109

Also a recent Italian study “TSUNAMI” did not show a plasma benefit in terms of reducing the risk of respiratory worsening or death in the first thirty days.

When speaking about “off label” approaches being evaluated worldwide, besides Tocilizumab and mesenchymal cells, also other treatments deserve to be mentioned. This article has an exhaustive table with current COVID-19 drugs under clinical investigation. DOI: 10.3892/ijo.2020.5159

94 Other therapies have not shown improvements in mortality 
95 or recovery in patients with COVID-19. In an RCT carried out in China, patients with 
96 severe or life-threatening COVID-19 symptoms were studied. The findings showed that 
97 convalescent plasma did not result in clinical improvement after 28 days of follow-up, 
98 demonstrating no superiority to standard medical management [27]. In India, these 99 findings were subsequently confirmed in a national multicenter RCT titled “PLACID” 
100 [28]. Lo que es consistente con el RCT “TSUNAMI”, publicado recientemente a partir de población italiana con COVID-19, en donde tampoco se evidenció que un título alto de anticuerpos presentes en plasma convaleciente, resultará en una reducción de la mortalidad a 30 días, ni menor necesidad de ventilación mecánica invasiva, señalando sólo un efecto marginal en ausencia de distress respiratorio agudo \cite{ItalianMedicinesAgency2020} . Other “off label” drugs are being evaluated worldwide to assess their usefulness 
101 against COVID-19. Tocilizumab, an IL-6 receptor antagonist, is being tested in an RCT 
102 (EudraCT: No. 2020-001408-41) in Germany to evaluate its efficacy and safety in patients 
103 with severe COVID-19 pneumonia. In the phase III RCT results, the use of Canakinumab, an IL-1β inhibitor, evidence to reduce the days of the hospitalization and mortality, in patients who do not require invasive ventilatory assistance, but do require supplemental O2 \cite{landi2020blockage}. On the other hand, the use of Sarilumab, an IL-6 receptor antagonist, in patients with severity criteria, no benefit was observed compared to placebo \cite{lescure2021sarilumab}. Other phase II RCTs involving Thalidomide (NCT04273529), and monoclonal antibodies such as Anakinra (NCT04603742), Gimsilumab (NCT04351243), and Ruxolitinib are ongoing (NCT04359290). 

LINES 123-124

The author states that NP develops a memory response from TLs that remain up to 11 years after infection. I think this sentence needs to be better clarified. After SARS-CoV infection (not SARS-CoV-2) it has been demonstrated that all memory T cells detected were specific against SARS-CoV structural S, N and M proteins (not only NP).  
Here is the original article for a better understanding.

122 or viral vectors[33,34]. Nucleocapsid protein (NP) was found to be expressed in large 
123 amounts during infection. NP is highly immunogenic and is a potential vaccine target. 
124 It develops a memory response from TLs that remain up to 11 years after SARS-CoV infection, detecting at the same time TLs of a specific memory, in addition to NP towards other structural proteins such as SP and MG \cite{ng2016memory}.

LINE 136 Typing error: change “o” with “or”.

135 led to a thickening of the pulmonary epithelium and severe pneumonia when using NP
136 or SARS-CoV but not SP [38]. Using NP as a vaccine target for SARS-CoV-2 is considered 
137 less safe when compared to SP.

LINES 136-137 Why using NP as a vaccine target for SARS-CoV-2 is considered less safe when compared to SP? I think the author should cite a source, here.

 The following citation was added to the document:
Yasui F, Kai C, Kitabatake M, Inoue S, Yoneda M, Yokochi S, Kase R, Sekiguchi S, Morita K, Hishima T, Suzuki H, Karamatsu K, Yasutomi Y, Shida H, Kidokoro M, Mizuno K, Matsushima K, Kohara M. Prior immunization with severe acute respiratory syndrome (SARS)-associated coronavirus (SARS-CoV) nucleocapsid protein causes severe pneumonia in mice infected with SARS-CoV. J Immunol. 2008 Nov 1;181(9):6337-48. doi: 10.4049/jimmunol.181.9.6337. PMID: 18941225.

Format:

LINE 136 Typing error: delete “however,”.

136 o SARS-CoV but not SP [38]. Using NP as a vaccine target for SARS-CoV-2 is considered 
137 less safe when compared to SP.

LINES 435-444

I think the fact that – of the 92 isolates of SARS-CoV-2 - only one came from LATAM, should be mentioned as a limitation of this study.

906 with loss of protective capacity[126]. Although we studied the most frequent alleles in 
907 LATAM, we only have one SARS-CoV-2 proteome from Brazil in the sample of proteomes (this can be considered as a limitation of the study, because of the limited amount of data to date),
908 that were selected for the in silico analysis described in the pipeline (Fig1- Pipeline) 
909 and available in (Table S1). We used a conservative approach by Eliminating the non
910 synonymous amino acid substitutions presented in the 92 proteomes collected between
911 December 2019 and March 11, 2020, from other regions of the world, mainly Asia

Reviewer 2 Report

In this article, the authors design by an immunoinformatics approach a candidate vaccine against SARS-COV-2 particularly suited for LATAM population. After analyzing the prevalence of HLA alleles in different LATAM countries, they selected peptides from SARS-COV-2 in non-structural and structural proteins in order to maximize the immune response from innate immune system and lymphocytes B and T. The selected peptides are fused in a protein harboring carrier adjuvant proteins. The several steps of the pipeline used are extensively described and the choice of peptide has been carefully argued and verified. Though the overall quality and merit of the manuscript is good, there are several misspelling to be corrected.

  1. In introduction, the paragraph about COVID19 treatment with Remdesivir, should be updated with latest publication. This field is still controversial but the WHO has stated that Remdesivir has limited therapeutic effects.
  2. In page 4 of Introduction (line 137-138) the authors wrote that NP is a less safe target than SP, though they use NP hereafter. This confusing sentence should be turned in another way.
  3. Though the authors mentioned SARS6COV-2 variants, how does their vaccine candidate will behave towards such variants? Multiepitope vaccines may have some adavantages to circumvent the “Original antigenic sin” problem with variants. The authors shoud discuss this point.
  4. The authors have modelled interaction of multiepitope with TLR4. Is there some common features with the modelling by Choudhury and Mukherjee (DOI: 10.1002/jmv.25987 )?

Minor spelling and typo errors

- page 4, line 143 “however,, however”

- page 9, line 369 “sum PDBS” for PDBsums

-page 9, line 374, Force field ff14SB43 parameters

-page 9 line 395 trajectories would be better than “paths”

-legend of Figure 2, “TRL” instead of TLR

-page 22 line 904, there is a repetition of “vaccine formulation”

Author Response

We are very grateful for the comments made on the manuscript, here you will find the answers to your concerns:

Corrections are made in blue color, as was changed in the main document

In the introduction, the paragraph about COVID19 treatment with Remdesivir should be updated with the latest publication. This field is still controversial but the WHO has stated that Remdesivir has limited therapeutic effects.

Currently, only Dexamethasone and Remdesivir have been shown to modify the 
84 natural course of the disease. The “RECOVERY” randomized clinical trial (RCT) was 
85 carried out in England. In this study, oral or intravenous Dexamethasone 6 mg/day was 
86 associated with a reduction in the mortality of patients requiring assisted ventilation, 
87 and patients who were in the second week of clinical diagnosis [24]. Another 
88 study, “ACTT1”, involved 10 countries in a multicenter RCT. The trial showed that the 
89 nucleotide analog Remdesivir, which used a loading dose of 200 mg and a maintenance 
90 dose of 100 mg/day, significantly reduced the time it took for patients to recover from 
91 COVID-19. Additionally, the need for supplemental oxygen after 10 days of intravenous 
92 treatment diminished [25]. However, the results of the WHO multicenter RCT “SOLIDARITY” study on the evaluation of the effect of “off label” drugs, which included the follow-up of about 11,330 adults from 4 continents, 30 countries and 405 hospitals, did not find Remdesivir to be associated with substantial prevention of in-hospital mortality, benefiting only a small fraction of patients when using a loading dose and standard maintenance \cite{who2021repurposed}.
92 Currently, the Food and Drug Association has approved 
93 the use of Remdesivir and Dexamethasone for COVID-19, although the latter is only a 
94 temporary approval

In page 4 of Introduction (line 137-138) the authors wrote that NP is a less safe target than SP, though they use NP here after. This confusing sentence should be turned in another way.

135 led to a thickening of the pulmonary epithelium and severe pneumonia when using NP
136 of SARS-CoV but not SP [38]. Therefore, the full-length protein NP as a vaccine target for SARS-CoV-2 could be considered less safe when compared to SP, since they share a homology greater than 90% \cite{tilocca2020comparative}. Consequently, if it is used as a vaccine target, shorter length approaches should be chosen, since immunogenic peptides of NP have been found capable of evoking an immune response and inhibiting viral replication in influenza A and SARS-COV viruses \cite{tilocca2020comparative, wang2003assessment, jiang2011inhibition, wei2020apoferritin, oliveira2020immunoinformatic}. 

Though the authors mentioned SARS-Cs deOV-2 variants, how does their vaccine candidate will behave towards such variants? / Multiepitope vaccines may have some adavantages to circumvent the “Original antigenic sin” problem with variants. The authors shoud discuss this point.

902 an important immunogenic region that is also associated with the linear epitopes of BLs.
903 However, it is important to keep in mind that using the complete SP subunit in a vaccine
904 formulation could be more effective in a fast vaccine formulation. It should be noted that
905 mutations in this immunodominant region could imply changes in antigenic recognition
906 with loss of protective capacity[126]. Our multi-epitope construct offers a specific immune response with greater attention to stimulating innate immunity, especially against relevant proteins in the phase of infection. Reflecting on whether it can offer cellular protection, even if antigenic drift or natural selection or by any SARS-CoV-2 vaccine itself throughout the pandemic or after, it may exceed the protection offered by neutralizing antibodies against the RBD. Protection that is not expected at least directly, when using the recombinant SP protein present in current commercial formulations obtained from the Wuhan-Hu-1 sequence \cite{williams2021sars}. Multi-epitope constructs, including the one proposed here, if required, can be periodically updated specifically covering new conserved B or T cell epitopes that emerge in SARS-CoV-2 variants, avoiding a delayed elimination of infected cells. Since by not using all the SP protein, other epitopes of SP present in most variants can be raised, without falling into the effect of the original antigenic sin \cite{chakradhar2015updated, tavasolian2021hla, williams2021sars}.  Although we studied the most frequent alleles in 907 LATAM, we only have one SARS-CoV-2 proteome from Brazil in the sample of proteomes 908 that were selected for the in silico analysis described in the pipeline (Fig1- Pipeline) 909 and available in (Table S1). We used a conservative approach by eliminating the non910 synonymous amino acid substitutions presented in the 92 proteomes collected between

The authors have modelled interaction of multiepitope with TLR4. Is there some common features with the modelling by Choudhury and Mukherjee (DOI: 10.1002/jmv.25987 )?
Minor spelling and typo errors

998 of its ectodomain, identifying a probable non-canonical interaction. At this location,
999 some P6 residues of MG have more contact near the more hydrophobic region of MD-2,
1000 where coupling with LPS has been reported[154]. Therefore, this interaction could be
1001 stable and safe since it shows the hydrophobic pocket of empty MD2 which is known to
1002 interact with traces of LPS and is associated with a fatal toxic immune response[155]. Otras interacciones proteicas que ocupan TLR-4 podrían ser relevantes en cuanto a seguridad. En COVID-19 se ha descrito un proceso inmunopatológico que al menos de forma teórica podría explicarse en parte a partir de la interacción de SP con TLR-4, incentivando la producción de citoquinas inflamatorias secundarias a la formación de interfaces compuestas por residuos antigénicos de SP interactuando con residuos externos de TLR-4 \cite{choudhury2020silico}. En nuestro constructo multie-pitope aunque usamos péptidos antigénicos que provienen de SP, ninguno de estos residuos antigénicos hacen parte de las interfaces descritas. In
1003 nature, other non-canonical interactions have been described as in the case of the HIV Tat
1004 protein that results in an interface between the N-terminal and TLR-4 / MD-2, showing
1005 greater stability in the resulting complex[156]. In addition, other studies on larger and
1006 smaller viral proteins, including Tat [157,158], 

- page 4, line 143 “however,, however”

141 been observed in patients with severe COVID-19 and convalescent patients [40,41]. NAb 
142 of NP has been related to the number of specific T cells which produce INF-γ [41]. In 
143 SARS-CoV, NP, MG, and E have been shown not to produce NAb; however, 
144 they are potential antigens for antiviral cytotoxic T cells (CTL) [42].

- page 9, line 369 “sum PDBS” for PDBsums

368 database; ID: 3FXI), and the refined multi-epitope construct was used as a ligand. The 
369 residues at the binding interface of the resulting complex were analyzed by PDBsum
370 and plotted by UCSF Chimera v1.14[77].

-page 9, line 374, Force field ff14SB43 parameters

-page 9 line 395 trajectories would be better than “paths”

. 394 Finally, 68 ns of production was simulated in an NPT assembly with a target 
395 pressure of 1 bar and a pressure coupling constant of 2 ps. Production trajectories were 
396 analyzed for 2 ps of the simulation using CPPTRAJ and PTRAJ

-legend of Figure 2, “TRL” instead of TLR

Figure 2. Antigenic presentation from the proposed vaccine: The assembly of peptides with vaccine potential are recognised by cell membrane receptors capable of recognising patterns associated with pathogens, such as TLR2 and 4 from dendritic cells. After recognition, the construct is phagocytised by the cell together with TLR, allowing its interaction with MyD88 and the maturation of the phagosome. From the phagosome, the peptide can take two routes: the first route is towards the proteasome, where the peptide is degraded, internalised in the ER and assembled with HLA-I molecules; the second route involves the internalisation of the peptide in the late endosome, where it is assembled with HLA-II molecules and subsequently presented on the cell membrane of the LT[

-page 22 line 904, there is a repetition of “vaccine formulation”

902 an important immunogenic region that is also associated with the linear epitopes of BLs. 
903 However, it is important to keep in mind that using the complete SP subunit 
904 could be more effective in a vaccine formulation. It should be noted that 
905 mutations in this immunodominant region could imply changes in antigenic recognition

Reviewer 3 Report

This paper covers a reverse engineering approach to design a multi-epitope vaccine for SARS-CoV-2 based on the identification of PPVPs from the conserved regions of proteins in 92 SARS-CoV-2 proteomes.

Searching for a better  vaccine which surges as the most cost-effective strategy for preventing infection and reducing COVID-19-related morbidity and mortality.

In the work,  proteomes of 92 SARS-CoV-2 strains were investigated and taken  from the Assembly database available at the National Center for Biotechnology Information (NCBI). Authors identified the most frequent HLA-I and HLA-II alleles in the LATin AMerica population, to predict potential epitopes for CTLs. They have predicted peptides related to HLA molecules most common in LATAM, the sequences of proteomes with at least 9 mers. Strenght: This is a very large and interesting covid paper, with Good figures and hard work. they came up with a candidate multi-epitope vaccine directed towards the world population, with an immunogenic capacity that according to authors, surpassed safety standards required to begin complementary experimental studies.

Weakness: please chech some tipos in english language.

Author Response

We are very grateful for the comments made on the manuscript, all tipos in english language where fixed.

This paper covers a reverse engineering approach to design a multi-epitope vaccine for SARS-CoV-2 based on the identification of PPVPs from the conserved regions of proteins in 92 SARS-CoV-2 proteomes.
Searching for a better  vaccine which surges as the most cost-effective strategy for preventing infection and reducing COVID-19-related morbidity and mortality.
In the work,  proteomes of 92 SARS-CoV-2 strains were investigated and taken  from the Assembly database available at the National Center for Biotechnology Information (NCBI). Authors identified the most frequent HLA-I and HLA-II alleles in the LATin AMerica population, to predict potential epitopes for CTLs. They have predicted peptides related to HLA molecules most common in LATAM, the sequences of proteomes with at least 9 mers. Strenght: This is a very large and interesting covid paper, with Good figures and hard work. they came up with a candidate multi-epitope vaccine directed towards the world population, with an immunogenic capacity that according to authors, surpassed safety standards required to begin complementary experimental studies.
Weakness: please check some tipos in english language.

Round 2

Reviewer 1 Report

a paragraph of the reply is in spanish languge

revision is accepted